| 1  | Measurement report: Observational insights into the impact of dust                                                                                                                    |
|----|---------------------------------------------------------------------------------------------------------------------------------------------------------------------------------------|
| 2  | transport on atmospheric dicarboxylic acids in ground region and free                                                                                                                 |
| 3  | troposphere                                                                                                                                                                           |
| 4  | Minxia Shen <sup>1</sup> , Weining Qi <sup>1</sup> , Yali Liu <sup>1,2</sup> , Yifan Zhang <sup>1,2</sup> , Wenting Dai <sup>1,3</sup> , Lu Li <sup>1</sup> , Xiao Guo <sup>1</sup> , |
| 5  | Yue Cao <sup>1,2</sup> , Yingkun Jiang <sup>1,2</sup> , Qian Wang <sup>1</sup> , Shicong Li <sup>1</sup> , Qiyuan Wang <sup>1,3</sup> , Jianjun Li <sup>1,3*</sup>                    |
| 6  |                                                                                                                                                                                       |
| 7  | <sup>1</sup> State Key Laboratory of Loess Science, Institute of Earth Environment, Chinese                                                                                           |
| 8  | Academy of Sciences, Xi'an 710061, China                                                                                                                                              |
| 9  | <sup>2</sup> Xi'an Institute for Innovative Earth Environment Research, Xi'an, China                                                                                                  |
| 10 | <sup>3</sup> National Observation and Research Station of Regional Ecological Environment Change                                                                                      |
| 11 | and Comprehensive Management in the Guanzhong Plain, Xi'an, China                                                                                                                     |
| 12 |                                                                                                                                                                                       |
| 13 |                                                                                                                                                                                       |
| 14 | *Corresponding author: Jianjun Li, e-mail address: <u>lijj@ieecas.cn</u>                                                                                                              |
| 15 |                                                                                                                                                                                       |

Abstract. Dust transport significantly affects downwind aerosol formation and regional 16 17 climate, yet the evolutionary mechanisms of SOA during this process remain poorly understood. Here, we conducted vertical observations of PM<sub>2.5</sub> and size-segregated aerosols at the foot and 18 19 top of Mount Hua, focusing on C<sub>2</sub> formation and its  $\delta^{13}$ C signatures influenced by dust transport. Under non-dust conditions, PM<sub>2.5</sub> and diacid concentrations at the foot were 4.5 and 2.1 times 20 higher than those at the top, indicating stronger anthropogenic influence at lower elevations. 21 Aerosols at the top revealed enhanced photochemical aging, with higher C<sub>2</sub>/C<sub>4</sub> (5.84 vs. 4.74), 22  $C_3/C_4$  ratios (1.04 vs. 0.56), and more positive  $\delta^{13}C$  values (-21.5% vs. -27.6%). The positive 23 correlation of C<sub>2</sub> with ALWC and its consistent size distribution with precursors confirm 24 aqueous-phase oxidation as the dominant formation pathway. During dust events, although 25 PM<sub>2.5</sub> concentrations increased, C<sub>2</sub> concentrations in PM<sub>2.5</sub> decreased by 59% at the foot and 26 25% at the top. Concurrently, the  $\delta^{13}$ C values of C<sub>2</sub> showed a positive shift, particularly at the 27 top (from -21.5% to -13.2%), suggesting that alkaline dust catalyzes the formation of <sup>13</sup>C-28 enriched oxalate. Size-segregated data revealed a shift of C2 from the fine to the coarse mode, 29 with the coarse-to-fine ratio increasing from 0.3–0.4 to 0.6–1.1. These findings demonstrate 30 that under dust influence, the primary formation pathway of C<sub>2</sub> shifts from aqueous-phase 31 oxidation in fine particles to heterogeneous reactions on coarse-particle surfaces. Moreover, 32 this shift is accompanied by a positive shift in the  $\delta^{13}$ C signature of C<sub>2</sub> and is more pronounced 33 34 at higher altitudes.

Keywords: Dicarboxylic acids, dust particles, size distribution, stable carbon isotopes ( $\delta^{13}$ C),

aqueous-phase oxidation, heterogeneous reactions

#### 1 Introduction

Dust cycling is crucial to Earth's climate system (Maher et al., 2010; Liang et al., 2022). Mineral particles from dust storms absorb and scatter solar radiation (Kumar et al., 2014), 39 altering regional heat balance and cloud properties, which in turn affects precipitation 40 (Mahowald et al., 2014; Kok et al., 2023; Marx et al., 2024; Xu-Yang et al., 2025). They 41 42 also act as effective ice-nucleating particles (INPs) in mixed-phase clouds, regulating ice formation and the radiation budget (Fan et al., 2016; Vergara-Temprado et al., 2018; Kawai, 43 et al., 2021; Chen et al., 2024). Dust particles often contain high levels of salts, bacteria, and 44 heavy metals, posing potential risks to human health and plant growth (Yamaguchi et al., 2016; 45 Luo et al., 2024). Dust not only impairs air quality locally but also undergoes long-range 46 transport, ultimately affecting both hemispheric and global climate systems (Pan et al., 2025). 47 During transport, mineral dust may undergo heterogeneous reactions, forming secondary 48 aerosols that aid cloud formation (Wang et al., 2020; Bikkina et al., 2023). The large surface 49 50 area of these particles facilitates reactions that alter radiation transfer and photolysis rates (Sullivan et al., 2007a). 51 Over the past 500 years, East Asia has been frequently hit by dust storms (Zhang et 52 53 al., 2021; Wu et al., 2022). The Taklimakan and Gobi Deserts, the primary sources of East Asian dust, emit over 800 million tons of dust to downwind areas annually (Sullivan et al., 54 2007b; Wang et al., 2015; Ren et al., 2019; Zhu and Liu, 2024). Northwestern and northern 55 China, frequently experience dusty weather due to these desert emissions (Gui et al., 2022; 56 57 Liang et al., 2022). In spring 2023, Mongolia contributed over 42% of the dust concentration in northern China (Chen et al., 2023). Although dust storms are more 58 common in China during spring (Sun et al., 2001), a large-scale, high-intensity dust storm 59 hit northern China on January 10, 2021. This severe dust storm, originating from southern 60 Mongolia and western Inner Mongolia, triggered rapid air quality deterioration across 61 downwind regions. Our synchronized field observations of PM<sub>2.5</sub> and size-segregated aerosols 62 63 at the top of Mount Hua and on the ground in the winter of 2021 successfully captured this large-scale dust event, as shown in Fig. S1, extensively covering Northern China and the 64

Guanzhong Plain. Liu et al. (2024) compared and analyzed the concentrations and size distributions of water-soluble inorganic ions during dust and non-dust periods, finding that the impact of dust on ground aerosols in the Guanzhong Plain is weaker than that in the free troposphere. Nevertheless, the specific mechanisms through which dust affects organic components, particularly secondary organic aerosol (SOA) and their precursors in the ground and troposphere remain unclear.

To investigate these processes, this study focuses on dicarboxylic acids (diacids), which serve as key tracers for SOA (Xu et al., 2022). As important components of water-soluble organic carbon, diacids are widely distributed in the atmosphere from the surface layer to the free troposphere (Fu et al., 2008; Myriokefalitakis et al., 2011). Conventional theory suggests that aqueous-phase chemical reactions occur predominantly in submicron particles containing water or cloud droplets (Lim et al., 2010; Ervens et al., 2011; Lamkaddam et al., 2021). However, field observations have reported the coexistence of oxalate and nitrate in supermicron particles during dust events (Falkovich and Schkolnik, 2004; Sullivan et al., 2007a; Wang et al., 2015; Xu et al., 2020). To explain this, Wang et al. (2015) proposed that the reaction of nitric acid and/or nitrogen oxides with dust generates (Ca(NO<sub>3</sub>)<sub>2</sub>), which absorbs water vapor to form an aqueous phase on the dust surface. This enables the partitioning of gas-phase watersoluble organic precursors into this aqueous phase, followed by their further oxidation to form oxalic acid (C<sub>2</sub>). Research by Li et al. (2025) provides direct evidence for this mechanism, showing that aqueous nitrate coatings (Ca(NO<sub>3</sub>)<sub>2</sub>), due to their very low deliquescence relative humidity (absorbing water at atmospheric RH > 8%), effectively promote the formation of aqueous secondary organic aerosols (aqSOA). Thus, aged dust surfaces provide critical reactive interfaces for aqSOA formation.

Tropospheric aerosols in high mountain areas are significantly influenced by long-range transport of surface pollutants, making them more representative of regional atmospheric quality. Our previous study (Shen et al., 2023) demonstrated that summer daytime valley winds on Mount Hua transport organic acids from the foot to top, thereby altering the chemical composition of the free troposphere and establishing distinct formation pathways of C<sub>2</sub> at

different altitudes. This study examines aerosol vertical distribution characteristics in winter. Low temperatures cause a significant reduction in the boundary layer height over Mount Hua, which inhibits the diffusion of local pollutants to the top. Consequently, the top remains in a free tropospheric environment where aerosols originate primarily from long-range transport from dust source regions (Liu et al., 2024). During the observational period, we documented a major dust event in which particles from dust source regions were directly transported to the top. There, they mixed with local anthropogenic pollutants, triggering complex atmospheric chemical reactions that resulted in notable vertical differences in aerosol chemical properties. However, the influence of heterogeneous reactions on dust aerosol surfaces on the generation of organic acids, particularly their role in modifying C<sub>2</sub> formation mechanisms at different altitudes remains unclear. Therefore, using observational data from a typical dust event in winter 2021, this study focuses on examining the impacts of dust transport on the molecular distribution, particle size characteristics of diacids, and the formation mechanisms of C<sub>2</sub>. The work aims to elucidate the key role of heterogeneous chemical reactions on the surfaces of dust aerosols in the formation of SOA, providing new insights into regional atmospheric chemical processes during dust events.

# 2 Experimental and methods

# 2.1 Sample collection

Samples were collected simultaneously at the free troposphere and the ground surface during December 17, 2020 to January 12, 2021. The sampling site at the ground surface is located on Yinquan Road, Huayin City, Weinan (34°31'N, 110°04'E; ~402 m a.s.l) (referred to as "Foot"), while alpine sampling site is located at the summit of the west peak of Mount Hua (34°28'N, 110°05'E; ~2065 m a.s.l) (referred to as "Top") (Fig. 1). PM<sub>2.5</sub> aerosol samples was collected using medium-flow sampler (HC-1010, China Qingdao Company, China) at a flow rate of 100 L min<sup>-1</sup> with a duration of 11 hours for each sample during the day (from 08:00 to 19:00) and night (from 20:00 to 07:00 the next day). A total of 54 samples were collected at both the alpine region and ground. The size-segregated samples were collected for ~71 h in each set using an Andersen multi-stage impactor (Andersen, Thermo electronic, USA) at a flow

- rate of 28.3 L min<sup>-1</sup> with 9 size bins as < 0.4, 0.4-0.7, 0.7-1.1, 1.1-2.1, 2.1-3.3, 3.3-4.7, 4.7-1.1
- 5.8, 5.8–9.0 and > 9.0  $\mu$ m, respectively. In a total of 9 sets of size-segregated samples were
- 123 collected. All the samples were collected onto pre-combusted (450°C for 6 h) quartz fiber filters
- produced by Whatman, UK. After sampling, the filters were stored in -18°C until analysis.

# 125 **2.2 Laboratory Analysis**

126

#### 2.2.1 Determination of carbonaceous species and water-soluble inorganic ions

- The concentrations of organic carbon (OC) and elemental carbon (EC) in PM<sub>2.5</sub> were
- determined using a DRI Model 2001 carbon analyzer (Atmoslytic Inc., USA), following the
- IMPROVE thermal/optical reflectance (TOR) protocol (Cao et al., 2007). A 0.526 cm<sup>2</sup> filter
- punch was heated stepwise in pure helium (at 120 °C, 250 °C, 450 °C, and 550 °C) followed
- by heating in a 2% oxygen/helium atmosphere (at 550 °C, 700 °C, 800 °C). The method
- detection limits were 0.41  $\mu$ g cm<sup>-2</sup> for OC and 0.03  $\mu$ g cm<sup>-2</sup> for EC.
- Water-soluble components were extracted from a quarter of each filter using 40 mL of
- ultrapure water (Milli-Q, 18.2 MΩ, Merck, France) via a combined process of 1-hour
- ultrasonication and 1-hour mechanical shaking. After filtration through a 0.45 µm membrane,
- the extracts were preserved at 4 °C for subsequent analysis. Water-soluble inorganic ions (NH<sub>4</sub><sup>+</sup>,
- K<sup>+</sup>, Mg<sup>2+</sup>, Ca<sup>2+</sup>, Cl<sup>-</sup>, NO<sub>3</sub><sup>-</sup>, and SO<sub>4</sub><sup>2-</sup>) were analyzed by ion chromatography (Metrohm 940,
- Switzerland). Anions and cations were separated using an IonPac AS23 column and an IonPac
- CS12A column, with 9.0 mM Na<sub>2</sub>CO<sub>3</sub> and 20 mM methanesulfonic acid as eluents,
- respectively (Zhang et al., 2011). Concurrently, water-soluble organic carbon (WSOC) was
- determined using a total organic carbon (TOC) analyzer (TOC-L CPH, Shimadzu, Japan) (Li
- et al., 2019). The detection limits for inorganic ions ranged from 0.008 to 0.022 µg m<sup>-3</sup>, while
- those for total carbon (TC) and inorganic carbon (IC) were 0.07 mg  $L^{-1}$  and 0.08 mg  $L^{-1}$ ,
- respectively.
- Aerosol liquid water content (ALWC) was calculated using the ISORROPIA-II model
- (Fountoukis and Nenes, 2007), based on the concentrations of water-soluble inorganic ions and
- meteorological parameters including relative humidity (RH) and temperature (T).
- Meteorological data for the top and the foot were obtained from the Mount Hua Meteorological

Station and the Huayin Meteorological Bureau, respectively. All statistical analyses were performed using SPSS.

# 2.2.2 Determination of dicarboxylic acids and related compounds

The analysis of diacids, keto-carboxylic acids, and α-dicarbonyls in PM<sub>2.5</sub> and sizesegregated aerosols was conducted based on an established derivatization method (Shen et al., 2022). A quarter of each filter was extracted by ultrasonicating in ultrapure water (Milli-Q, 18.2 M $\Omega$ , Merck, France) for three sequential 15-minute intervals. To maximize the recovery of low-molecular-weight acids (e.g., C<sub>2</sub>), the extracts were alkalized to pH 8.5–9.0 with 0.1 M KOH prior to the concentration step. The aqueous extracts were concentrated to near-dryness under vacuum in a water bath maintained at 55 °C (evaporation was halted immediately after the disappearance of the last drop of solvent). The dried residues were derivatized with 14% BF<sub>3</sub>/n-butanol at 100°C for 1 hour to convert carboxyl groups into dibutyl esters and oxo groups into dibutoxyacetals. After the reaction, the derivatives were sequentially dissolved in n-hexane, acetonitrile, and pure water, followed by triple extraction via vortex mixing to remove watersoluble inorganic substances. The aqueous lower layer was removed using a Pasteur pipette. The n-hexane layer was concentrated by rotary evaporation and N<sub>2</sub> blow-down, reconstituted in 100 µL n-hexane, and finally analyzed by GC-FID (HP 6890, Agilent Technologies, USA) (Wang et al., 2012). The recovery of the target compounds was 83% for C2 and ranged from 87% to 110% for other diacids. Stable carbon isotopic compositions ( $\delta^{13}$ C) of C<sub>2</sub> were determined using gas chromatography-isotope ratio mass spectrometry (GC-IRMS; Delta V Advantage, Thermo Fisher Scientific, Franklin, MA, USA) following established protocols (Kawamura and Watanabe, 2004). To ensure analytical precision (standard deviation < 0.2%), derivatized samples were analyzed in triplicate. Final  $\delta^{13}$ C values of free oxalic acid were calculated via mass-balance correction to account for isotopic contributions from the BF<sub>3</sub>/n-butanol derivatizing agent.

#### 3 Results and discussion

# 3.1 Vertical differences in PM<sub>2.5</sub> chemical composition at the foot and top of Mount Hua

# during non-dust periods

During non-dust periods, significant vertical differences were observed in the chemical composition of PM<sub>2.5</sub> between the foot and top of Mount Hua. The average PM<sub>2.5</sub> concentration at the foot  $(127 \pm 48 \ \mu g \ m^{-3})$  was 4.5 times higher than that at the top  $(28 \pm 14 \ \mu g \ m^{-3})$ , with a mean difference of 99  $\mu$ g m<sup>-3</sup> (95% CI: 86 to 113; t (61.05) = 14.60, p < 0.001) (Supplementary Table S1 and Table S3a). Carbonaceous aerosols and water-soluble ions were significantly enriched at the foot (all p < 0.001; Table S3a). As key components of WSOC (Kawamura et al., 2016; Yang et al., 2020), the total concentration of diacids at the foot ( $1808 \pm 1280 \text{ ng m}^{-3}$ ) was approximately 2.1 times higher than those at the top (860  $\pm$  534 ng m<sup>-3</sup>), with a mean difference of 718 ng m<sup>-3</sup> (95% CI: 453 to 983; t (69.15) = 5.40, p < 0.001; Table S3b). The spatial difference was most pronounced for azelaic acid (C<sub>9</sub>), a biomarker of biomass burning (Kalogridis et al., 2018; Shen et al., 2022), with the concentration at the foot (153  $\pm$  110 ng  $m^{-3}$ ) being 5.7 times higher than that at the top (27 ± 18 ng  $m^{-3}$ ), corresponding to a mean difference of 126 ng m<sup>-3</sup> (95% CI: 95 to 156; t (54.77) = 8.24, p < 0.001; Table S3b). Consequently, the contribution of C<sub>9</sub> to total diacids was substantially greater at the foot (8.5%) than at the top (3.2%) (Fig. 2b). Correlation analysis of PM<sub>2.5</sub> and its chemical components revealed no significant relationship between the two sites during winter, contrasting with the positive correlations observed in summer (Shen et al., 2023). Combined with ion composition and back-trajectory analysis from Liu et al. (2024), these findings indicate that pollutants at the top primarily originate from regional transport from the northwestern direction with minimal vertical mixing, while components at the foot of the mountain were mainly controlled by local emission sources. Day-night differences provide further evidence supporting this conclusion. At the foot of the mountain, PM<sub>2.5</sub>, ionic components, and carbonaceous components (except for NO<sub>3</sub><sup>-</sup> and NH<sub>4</sub><sup>+</sup>) all showed significant day-night differences (p < 0.05; Table S4a). Among the diacids, methylglyoxal (mGly) exhibited the most pronounced day-night differences (daytime: 128 ± 49 ng m<sup>-3</sup>, 47% higher than nighttime:  $87 \pm 42$  ng m<sup>-3</sup>, p < 0.001; Fig. S2, Table S4b). As the terminal product of diacids photo-oxidation (Kawamura and Sakaguchi, 1999), C2 also

displayed marked day-night differences (daytime:  $766 \pm 552$  ng m<sup>-3</sup> vs. nighttime:  $585 \pm 497$ ng  $m^{-3}$ , p = 0.023), reflecting strong anthropogenic influence on ground-level photochemistry. In contrast,  $C_2$  at the top of the Mount Hua (daytime:  $312 \pm 224$  ng m<sup>-3</sup> vs. nighttime:  $299 \pm$ 186 ng m<sup>-3</sup>; p = 0.941) and its precursors showed no clear day-night differences (p = 0.341– 0.917; Table S4c), consistent with the patterns of PM<sub>2.5</sub> (p = 0.979), OC (p = 0.766), and other major components (Table S4a). Such stability is a typical characteristic of high-altitude sites located above the planetary boundary layer, primarily governed by regional transport processes (Fu et al., 2008; Li et al., 2012; Meng et al., 2014), indicating that aerosol processes in the free troposphere differ from those at ground level. In this high-altitude environment, in-cloud processes represent a key pathway for agSOA formation. Studies have shown that C<sub>2</sub> mainly originates from the in-cloud oxidation of precursors such as glyoxal (Gly) and isoprene (Warneck, 2003; Lim et al., 2005; Carlton et al., 2006), a mechanism supported by global model simulations (Myriokefalitakis et al., 2011). Additionally, the photochemical decomposition of C<sub>2</sub> following its association with Fe-containing particles in clouds (Zhang et al., 2019) also contributes to the stable distribution of organic acid concentrations at high altitudes.

Aerosol aging indicators further confirmed distinct oxidation processes between different altitudes. According to existing research, succinic acid (C<sub>4</sub>) can be via hydroxylation to generate C<sub>2</sub> and malonic acid (C<sub>3</sub>), while C<sub>3</sub> can be further converted to C<sub>2</sub> through intermediates such as hydroxymalonic acid or ketomalonic acid (Kawamura et al. 1993; Kunwar and Kawamura, 2014; Hoque et al., 2017). Therefore, the C<sub>2</sub>/C<sub>4</sub> and C<sub>3</sub>/C<sub>4</sub> ratios are widely used as effective indicators for assessing the extent of photochemical aging in organic aerosols (Kawamura et al., 2016; Meng et al., 2018; Shen et al., 2022). In this study, the C<sub>2</sub>/C<sub>4</sub> ratio at the top of Mount Hua was  $5.84 \pm 0.32$ , and the C<sub>3</sub>/C<sub>4</sub> ratio was  $1.04 \pm 0.08$ , both higher than the corresponding values at the foot (4.74  $\pm$  0.28 and 0.56  $\pm$  0.05, respectively; Fig. 2c). These results are consistent with reports from high-altitude areas such as Mount Tai (Wang et al., 2009; Meng et al., 2018) and Mount Hua (Meng et al., 2014), collectively confirming that

the atmosphere undergoes more significant photochemical aging during long-range transport at high altitudes.

This study compared the correlations between  $C_2$  and its key precursors at the foot and top of Mount Hua (Fig. 3a–d, 3i–l), revealing differences in atmospheric oxidation pathways across altitudes. Building on this, the synergistic effects of inorganic ions and ALWC on  $C_2$  formation were investigated (Fig. 3e–h, 3m–p), thereby elucidating the aqueous-phase reaction mechanisms and the role of environmental factors such as humidity in  $C_2$  generation. Results showed that  $C_2$  exhibited the strongest correlation with glycolic acid ( $\omega C_2$ ) ( $R^2 = 0.88$  at the foot,  $R^2 = 0.95$  at the top, p < 0.01; Fig. 3 d, l). Other key precursors included Gly ( $R^2 = 0.57$ -0.67, P < 0.01; Fig. 3b, 3j), mGly ( $R^2 = 0.58$ -0.85, P < 0.01; Fig. 3a, 3i), and pyruvic acid (Pyr:  $R^2 = 0.80$ -0.81, P < 0.01; Fig. 3c, 3k). These correlation characteristics confirm that aqueous-phase oxidation serves as the primary formation pathway for  $C_2$  under non-dust conditions (Deshmukh et al., 2017; Du et al., 2022). The consistently higher correlation coefficients at the top further substantiate the enhancing effect of prolonged atmospheric processes on secondary organic aerosol formation.

Contrary to previous studies (Wang et al., 2012; Meng et al., 2018), we found that  $C_2$  correlated more strongly with  $NO_3^-$  ( $R^2 = 0.79$ , p < 0.01; Fig. 3f) than with  $SO_4^{2-}$  ( $R^2 = 0.71$ , p < 0.01; Fig. 3e) at the foot. The difference mainly stems from strong influences of local anthropogenic emissions (e.g., traffic and industrial activities) at the foot, which provide abundant  $NO_x$  and lead to an increased proportion of  $NO_3^-$  in  $PM_{2.5}$ . Since  $NO_3^-$  is more hygroscopic than  $SO_4^{2-}$ , its elevated concentration further enhances ALWC, thereby promoting aqSOA formation and intensifying heterogeneous reaction processes (Huang et al., 2025). In contrast, at the top of Mount Hua,  $C_2$  exhibited a higher correlation with  $SO_4^{2-}$  ( $R^2 = 0.63$ , p < 0.01; Fig. 3m) than with  $NO_3^-$  ( $R^2 = 0.37$ , p < 0.01; Fig. 3n). This phenomenon is closely related to the active in-cloud processes in the free troposphere mentioned above.  $SO_4^{2-}$  at the top primarily originates from the oxidation of  $SO_2$  within cloud droplets (Yermakov et al., 2023), a process dominated by heterogeneous reactions (Wang et al., 2025). Meanwhile, cloud droplets also provide key reaction media for the aqueous-phase photooxidation of  $C_2$  precursors

such as Gly and  $\omega C_2$  (Warneck, 2003).

ALWC exhibited a humidity-dependent effect on  $C_2$  formation. Under RH<75%, ALWC was positively correlated with  $C_2$  concentration ( $R^2=0.36\text{-}0.44$ , p<0.01; Fig. 3g, 3o), reflecting the promotion of precursor dissolution and oxidation by the expansion of the aqueous phase. However, when RH exceeded 75%, supersaturation shifted the gas-particle partitioning equilibrium, causing  $C_2$  concentrations to decrease with increasing ALWC. After excluding high-humidity data, the correlation between ALWC and  $C_2$  significantly strengthened ( $R^2=0.59$ , p<0.01; Fig. 3h, 3p), confirming that aqueous-phase oxidation is the primary pathway for  $C_2$  formation during non-dust periods. This finding complements the research by Yang et al. (2022) on the synergistic effects of humidity and pH, jointly revealing the complex regulatory mechanisms of  $C_2$  formation under different humidity conditions.

# 3.2 Impact of dust transport on PM<sub>2.5</sub> chemical composition

On January 10, 2021, an extensive and intense dust storm, driven by successive cold fronts and sustained high-velocity winds, swept across northern China, triggering a dramatic surge in PM<sub>2.5</sub> concentrations (Fig. S3). At the foot of Mount Hua, PM<sub>2.5</sub> concentrations rapidly rose from 95  $\mu$ g m<sup>-3</sup> to 457  $\mu$ g m<sup>-3</sup> within 24 hours, reaching 3.4 times the non-dust average (127  $\pm$  48  $\mu$ g m<sup>-3</sup>). Concurrently at the top, PM<sub>2.5</sub> climbed from 46 to 165  $\mu$ g m<sup>-3</sup>, representing a 5.9-fold rise compared to typical conditions (28  $\pm$  14  $\mu$ g m<sup>-3</sup>). The cleaner atmospheric environment at the top amplified the relative impact of dust transport, while at the foot, existing local pollution partially masked the dust contribution. This contrast highlights the altitude-dependent response to dust events, with the top showing greater sensitivity due to its lower background pollution levels.

HYSPLIT trajectory analysis revealed the dust originated from the Inner Mongolia-Gansu arid region and was transported along a northwest path to the study area (Fig. S4). Dust transport was closely linked to atmospheric circulation, especially in the troposphere, where changes in wind speed play a key role in dust dispersion (Yang et al., 2017). During the dust period, wind speeds at the foot increased from 2.0 to 5.3 m s<sup>-1</sup>, and high-altitude wind speeds reached 12.2 m s<sup>-1</sup>, higher than the average wind speed during non-dust periods ( $5.4 \pm 3.0$  m

 $s^{-1}$ ).

Although PM<sub>2.5</sub> absolute concentrations rose during the dust period, component concentration changes at the two sites differed markedly. At the foot of the Mount Hua, EC concentrations remained relatively stable ( $4.5 \pm 2.1$  during non-dust vs.  $4.8 \pm 1.8$  µg m<sup>-3</sup> during dust; Table S2). OC increased from  $17 \pm 8.0$  to  $19 \pm 4.6$  µg m<sup>-3</sup>, but its mass fraction in PM<sub>2.5</sub> decreased substantially from 13.4% to 4.4%, reflecting the overwhelming contribution of mineral dust. The top exhibited different characteristics, with both EC and OC concentrations doubling from 0.9 to 1.8 µg m<sup>-3</sup> and 4.7 to 9.4 µg m<sup>-3</sup> respectively. OC maintained a higher mass fraction of 6.5% compared to 4.4% at the foot. These patterns indicate efficient mixing of dust with anthropogenic carbonaceous aerosols during long-range transport, coupled with more vigorous secondary formation processes in the free troposphere (Wang et al., 2023; Zheng et al., 2024).

Concentrations of mineral components like calcium and magnesium ions (Ca<sup>2+</sup> and Mg<sup>2+</sup>) rose (foot: 1.8 to 7.7 µg m<sup>-3</sup>; top: 0.7 to 3.2 µg m<sup>-3</sup>), confirming their established role as reliable tracers of dust emissions (Li et al., 2016; Liu et al., 2024). SO<sub>4</sub><sup>2-</sup> concentrations also increased at both sites (foot: 5.8 to 10.0 µg m<sup>-3</sup>; top: 3.8 to 8.7 µg m<sup>-3</sup>). This increase can be attributed to both the release of inherent sulfate species in dust (e.g., CaSO<sub>4</sub>) (Wu et al., 2012) and heterogeneous reactions on dust particle surfaces, where transition metals such as Fe (III) and Mn (II) catalyze the conversion of SO<sub>2</sub> to SO<sub>4</sub><sup>2-</sup> (Harris et al., 2013; Myriokefalitakis et al., 2022).

Dust transport altered the concentrations and molecular distribution of diacids and their precursors. Although  $C_2$  remained the most concentrated acidic molecule during dust periods, its absolute concentration decreased noticeably. At the foot of Mount Hua,  $C_2$  concentrations dropped from  $674 \pm 528$  ng m<sup>-3</sup> (non-dust periods) to  $276 \pm 20$  µg m<sup>-3</sup> (dust periods), decrease by 59%. At the top of Mount Hua,  $C_2$  concentrations decrease from 306  $\pm$  204 ng m<sup>-3</sup> to 229  $\pm$  45 ng m<sup>-3</sup>, a reduction of 25%. Severe ozone (O<sub>3</sub>) pollution was present in this dust storm event, and the particulate eruption promoted the generation and dispersion of O<sub>3</sub> pollutants. O<sub>3</sub> concentrations at the foot increased sharply from 15 µg m<sup>-3</sup> to 62 µg m<sup>-3</sup> (Fig.

S3), much higher than the non-dust average of  $26 \pm 19~\mu g~m^{-3}$ . The dust's extinction effect likely reduced aerosol optical thickness, enhancing surface UV radiation. Combined with a local temperature rise ( $\Delta T = +5.8^{\circ}C$ ), this probably triggered free-radical chain reactions, promoting the heterogeneous oxidation of  $C_2$  on mineral surfaces (Usher, et al., 2003; Lu et al., 2023). Notably, the proportion of  $C_2$  in total diacids exhibited contrasting trends at the two sites, decreasing from 37.3% to 32.2% at the foot and increasing from 35.5% to 42.8% at the top (Fig. 4c). This divergence could be closely related to the humidity levels at the two sites. During the dust event, the lower relative humidity at the foot (RH = 24 ± 8.5%) suppressed aqueous-phase oxidation, whereas the higher humidity at the top (RH = 44 ± 11%) favored  $C_2$  formation through such reactions. Additionally, variations in aerosol sources, transport pathways, aging processes, and potential contributions from other chemical reactions may also have influenced  $C_2$  generation.

As a major atmospheric keto acid and key precursor of C<sub>2</sub> (Kawamura et al., 2012; 2013), ωC<sub>2</sub> ranked second among the acids detected at both sites during non-dust periods, with concentrations of 214  $\pm$  199 ng m<sup>-3</sup> at the foot and 77  $\pm$  52 ng m<sup>-3</sup> at the top of Mount Hua. However, during dust periods, mGly became the second most abundant acid at both sites due to its enrichment on dust-particle surfaces. At the foot, mGly concentration reached  $116 \pm 34$ ng m<sup>-3</sup> (13.6% of total diacids), up from 5.9% in non-dust periods. At the top, mGly concentration was  $38 \pm 9$  ng m<sup>-3</sup> (7.1% of total diacids) (Fig. 4b), slightly higher than the nondust 5.6%. Phthalic acid (Ph), a photo-oxidation product of naphthalene and other aromatic hydrocarbons, primarily originates from industrial processes and incomplete combustion of coal in heavy and diesel vehicles (Ho et al., 2006). At both foot and top of Mount Hua, the proportion of Ph remains relatively stable, accounting for approximately 4% during both dust and non-dust periods, indicating that its sources are stable and closely related to regional industrial activities and traffic emissions. During dust periods, the C2/C4 ratios at the foot and top of Mount Hua rose to 5.24 and 7.75 (Fig. 4c), showing stronger aerosol aging. However, the C<sub>3</sub>/C<sub>4</sub> ratio at top of Mount Hua dropped to 0.70, this might result from the combined effects of selective adsorption of C2 onto dust particles and enhanced photolysis of C<sub>3</sub> on mineral dust surfaces.

# 3.3 Size distribution characteristics of diacids and related compounds during non-dust and dust periods

During non-dust periods, the size distribution of C<sub>2</sub> at both the foot and top of Mount Hua exhibited a distinct bimodal distribution (Fig. 5a and 5(a)), characterized by a primary peak in the fine particle mode (0.4-1.1 μm) and a secondary peak in the coarse particle mode (4.7-5.8 μm). Fine particles, with their greater specific surface area and hygroscopic nature, provide a conducive liquid-phase environment that enhances the oxidation of precursors such as mGly and Pyr, leading to C<sub>2</sub> formation (Ervens et al., 2011; Wang et al., 2015). The high correlation between C<sub>2</sub> and secondary inorganic ions (SO<sub>4</sub><sup>2-</sup>, NO<sub>3</sub><sup>-</sup>, NH<sub>4</sub><sup>+</sup>) (with R<sup>2</sup> values of 0.92-0.95 at the foot and 0.48-0.92 at the top, p <0.01; as shown in Fig. 6) supports this mechanism, confirming that C<sub>2</sub> formation during non-dust periods primarily relies on liquid-phase oxidation reactions on fine particle surfaces. The lower R<sup>2</sup> values at the top may be due to the greater influence of long-range transport at high-altitude sites, resulting in more complex sources of precursors. C<sub>2</sub> in the coarse particle mode likely originates from direct adsorption of biogenic emissions (such as plant waxes) or heterogeneous oxidation of gas-phase precursors on mineral dust surfaces (Wang et al., 2012).

The distribution patterns of short-chain diacids, such as C<sub>3</sub> and C<sub>4</sub> are similar to that of C<sub>2</sub> (Fig. 5b-c and 5(b)-(c)), with primary peaks at 0.4-1.1 μm and secondary peaks at 4.7-5.8 μm, indicating that these acids mainly come from fine particles. In contrast, glutaric (C<sub>5</sub>) acid shows distinct distribution characteristics at the two sites. At the foot, it exhibits a unimodal distribution in the coarse particles (4.7-5.8 μm) (Fig. 5d), while at the top, it displays a bimodal distribution (0.4-1.1 μm and 4.7-5.8 μm) (Fig. 5(d)). This significant difference in modal structure suggests that different atmospheric processes govern the behavior of C<sub>5</sub> at different altitudes. Adipic (C<sub>6</sub>) acid shows a bimodal distribution at both sites, probably resulting from the oxidation of cyclohexene (in the fine mode) or adsorption of gas-phase precursors on coarse particle surfaces (Deshmukh et al., 2016). Azelaic (C<sub>9</sub>) acid is enriched only in fine particles at the foot, which may be closely related to the oxidation of unsaturated fatty acids emitted from

biomass burning in northern regions during winter (Deshmukh et al., 2016). The coarse particle fraction of Ph likely forms through gas-phase adsorption, consistent with the strong adsorption characteristics of coarse particles reported by Kanellopoulos et al. (2021).

The particle size distributions of mGly (Fig. 5g and 5(g)) and  $\omega$ C<sub>2</sub> (Fig. 5j and 5(j)) showed remarkable consistency with that of C<sub>2</sub> (Fig. 5a and 5(a)), providing direct evidence that aqueous-phase oxidation serves as the predominant formation pathway for C<sub>2</sub>. Pyr exhibited distinct altitudinal variation in its distribution characteristics. At the top of Mount Hua, Pyr displayed a similar size distribution pattern to C<sub>2</sub>, indicating their shared photochemical origin. In contrast, at the foot, Pyr demonstrated enrichment in coarse particles (4.7-5.8 µm), likely attributable to heterogeneous reactions of gaseous precursors from local coal combustion on mineral dust surfaces. Gly also exhibits a peak in coarse particles, likely due to its strong adsorption and chemical stability on particle surfaces. The diacids concentrations are consistently lower at the top compared to the foot, which is closely tied to substantial local emissions at the lower elevation. As shown in Fig. 6, the coal combustion and biomass burning tracer Cl<sup>-</sup> demonstrates an exceptionally strong correlation with C<sub>2</sub> (R<sup>2</sup> = 0.88, p< 0.01) at the foot.

The dust transport process impacted the size distributions of diacids in aerosols (Fig. 5). At the foot of the mountain, the C<sub>2</sub> concentration in fine particles (≤2.1 μm) decreased from 8662 ng m<sup>-3</sup> to 4880 ng m<sup>-3</sup> (a reduction of 43.7%) during dust events, while the concentration in coarse particle (>2.1 μm) increased from 2718 ng m<sup>-3</sup> to 2843 ng m<sup>-3</sup> (an increase of 4.6%) (Table 3). A more pronounced change was observed at the high-altitude top of Mount Hua, where the C<sub>2</sub> concentration in coarse particles (2301 ng m<sup>-3</sup>) exceeded that in fine particles (2161 ng m<sup>-3</sup>) during dust events, indicating a shift in the dominant particle size distribution from fine to coarse modes. This shift in particle size distribution can be attributed to the formation of Ca(NO<sub>3</sub>)<sub>2</sub> coatings resulting from the reaction between calcium carbonate and NO<sub>3</sub><sup>-</sup> during dust aging (Li and Shao, 2009; Zhi et al., 2025). These hygroscopic coatings create favorable conditions for the adsorption and oxidation of gaseous organic compounds, thereby promoting the formation of SOA on the surfaces of coarse particles. Research by Li et al. (2025)

further confirms that aqSOA formed on dust surfaces can effectively enhance SOA production and drive a transition in the size distribution from the submicron to the supermicron range, which is highly consistent with the observational results of this study. Size-segregated ion data (Fig. S5) provide direct evidence for the above mechanism. Ca<sup>2+</sup> was primarily present in the coarse mode (3.3–5.8 μm), while during dust events, NO<sub>3</sub><sup>-</sup> at the top migrated from the fine mode (0.4–1.1 μm) to the coarse mode (3.3–5.8 μm) and coexisted with Ca<sup>2+</sup> in the same size range, strongly supporting the formation of Ca(NO<sub>3</sub>)<sub>2</sub> coatings on dust particle surfaces. In contrast, at the foot of the mountain, although the concentration of NO<sub>3</sub><sup>-</sup> decreased, it remained predominantly in the fine mode. This spatial difference may stem from more thorough aging and reactions of aerosols at the top due to longer transport times. Meanwhile, the foot is influenced by local pollution, resulting in higher background NO<sub>3</sub><sup>-</sup> concentrations and competitive reactions with components such as SO<sub>4</sub><sup>2-</sup>, which may collectively delay the distinct shift of NO<sub>3</sub><sup>-</sup> to the coarse mode.

Analysis of the dust/non-dust concentration ratio ( $R_{D/N}$ ) revealed  $R_{D/N}$  values of 0.3 and 0.6 for fine and coarse particles at the foot, respectively, while these values reached 0.4 and 1.1 at the top, indicating that dust processes have a more impact on the particle size distribution of diacids at high-altitude regions. Further investigations confirmed enrichment of  $C_2$  precursors (mGly, Pyr, and  $\omega C_2$ ) in the coarse particle fraction (>2.1  $\mu$ m). Observational data from the top of Mount Hua revealed that the concentration ratio ( $R_{D/N}$ ) of these precursors in coarse particles during dust periods reaches 0.8-1.4 during dust periods, higher than the 0.5-0.7 ratio in fine particles ( $\leq$ 2.1  $\mu$ m), demonstrating the crucial contribution of heterogeneous oxidation on dust particle surfaces to  $C_2$  formation. Throughout dust episodes, the particle size distribution patterns of diacids ( $C_2$ - $C_6$ ) consistently displayed a pronounced shift from fine to coarse particles (4.7-5.8  $\mu$ m) (Fig. 5). Notably, concentrations of the biomass burning tracer  $C_9$  decreased during dust episodes, while Ph and isophthalic acid (iPh) showed distinct peaks in the coarse particle mode. This phenomenon indicates that dust particles effectively scavenge gaseous pollutants through strong adsorption, thereby suppressing the formation of local fine-mode SOA. However, these gaseous precursors adsorbed onto coarse particle surfaces can still

undergo heterogeneous oxidation reactions to form SOA.

During dust events, the correlation between C<sub>2</sub> and mineral ions (Ca<sup>2+</sup>, Mg<sup>2+</sup>) showed significant enhancement (with R<sup>2</sup> values of 0.33, 0.30; p<0.05 at the foot and 0.65, 0.39; p<0.01 at the top). This finding showed excellent agreement with the recent research results of Li et al. (2025), who similarly observed stronger correlations between  $Ca^{2+}$  and  $C_2$  ( $R^2 = 0.46-0.95$ ) in the coarse particle phase. This arises as organic acids like C2 in aged carbonate-containing dust particles react with carbonates to form stable salts (Ervens et al., 2008; Lim et al., 2010). This process inhibits the volatility of organic acids and stabilizes them in the coarse particle phase. Furthermore, due to differences in rock types and weathering processes, Asian dust particles inherently contain higher concentrations of alkaline metal elements (Ca and Mg) compared to dust from other regions (Yu et al., 2025). The size-resolved correlations between C2, Ca2+, and Mg<sup>2+</sup> (Fig. 6), further support this conclusion. During dust events, the concentrations of C<sub>2</sub>, Ca<sup>2+</sup>, and Mg<sup>2+</sup> exhibited synchronous increases with increasing particle size. In contrast, during non-dust periods, the peak concentration of C2 occurred in the fine particle size range, while mineral ion concentrations did not show corresponding increases. As described in Section 3.2, dust events led to reductions in  $C_2$  concentrations in  $PM_{2.5}$  (foot:  $674 \pm 528$  ng m<sup>-3</sup> during non-dust periods vs.  $276 \pm 20$  ng m<sup>-3</sup> during dust periods; top:  $306 \pm 204$  ng m<sup>-3</sup> during non-dust periods vs.  $229 \pm 45$  ng m<sup>-3</sup> during dust periods).

Overall, the transformation mechanisms of C<sub>2</sub> and its precursors underwent alterations during dust events, shifting from aqueous-phase oxidation dominated in fine particles during non-dust periods to heterogeneous oxidation on coarse particle surfaces as the primary pathway during dust episodes. Regional comparative analysis further revealed that atmospheric chemical processes at the mountain foot were mainly influenced by local emission sources such as coal combustion and biomass burning, whereas the summit site more clearly reflected the complex interactions between long-range dust transport and regional atmospheric processes.

# 3.4 Stable carbon isotopes (δ<sup>13</sup>C) of oxalic acid

Synchronized observations at the foot and top of Mount Hua (2060 m asl) revealed an inverse correlation between  $C_2$  concentration and  $\delta^{13}C$  values in PM<sub>2.5</sub> (Fig. 7a and 7b). When

 $C_2$  concentration at the foot increased to 2424 ng m<sup>-3</sup>, its  $\delta^{13}$ C value decreased to -34.5‰, while at the top, a concentration of 917 ng m<sup>-3</sup> corresponded to a  $\delta^{13}$ C of -24.7‰. Conversely, during low concentration periods,  $\delta^{13}$ C at the foot rose to -21.9% (258 ng m<sup>-3</sup>) and at the top to -17.2% (63 ng m<sup>-3</sup>). This systematic variation provides clear evidence for kinetic carbon isotope fractionation during atmospheric aqueous-phase oxidation processes. Specifically, volatile organic compounds (VOCs) and semi-volatile organic compounds (SVOCs) with lower  $\delta^{13}C$ values preferentially react to form aqSOA, resulting in <sup>13</sup>C-depleted products (Xu et al., 2022). Spatially, the  $\delta^{13}$ C values of C<sub>2</sub> in PM<sub>2.5</sub> at the top of Mount Hua (-28.4% to -12.8%), mean -21.5%) were higher than those at the foot (-36.2% to -14.9%, mean -27.6%). This vertical gradient primarily results from long-range transport of aerosols at high altitudes coupled with deep oxidation processes. The higher C<sub>2</sub>/C<sub>4</sub> ratio observed at Mount Hua (5.84 vs. 4.74 at the foot) indicates more pronounced atmospheric aging characteristics. This distribution pattern originates from prolonged photochemical oxidation during long-range transport, where preferential cleavage of <sup>12</sup>C-<sup>12</sup>C bonds (due to their lower bond energy) leads to relative <sup>13</sup>C enrichment in residual C<sub>2</sub>. In contrast, surface aerosols dominated by local fresh emissions undergo shorter oxidation periods and exhibit weaker isotope fractionation effects. Moreover,  $\delta^{13}$ C can provide insights into the sources of aerosols, Pavuluri and Kawamura (2016) found that biogenic aerosols had higher mean  $\delta^{13}$ C values (-15.8‰) than anthropogenic sources (-19.5%). Our study shows that foot aerosols were mainly influenced by anthropogenic sources (biomass burning and coal combustion), while the top was more affected by natural sources due to richer vegetation, with long-range transport potentially weakening local isotope fractionation effects.

During dust events,  $C_2$  concentrations in  $PM_{2.5}$  showed decreasing trends, declining from  $674 \pm 528$  ng m<sup>-3</sup> to  $276 \pm 20$  ng m<sup>-3</sup> at the foot and from  $306 \pm 204$  ng m<sup>-3</sup> to  $229 \pm 45$  ng m<sup>-3</sup> at the top of Mount Hua. Concurrently, the  $\delta^{13}C$  values of  $C_2$  exhibited distinct positive shifts, increasing from -27.6‰ to -23.9‰ at the foot and from -21.5‰ to -13.2‰ at the top. This phenomenon reveals key chemical transformation mechanisms during dust transport: alkaline mineral surfaces promote heterogeneous catalytic oxidation of  $C_2$  precursors, with coarse-

mode mineral components ( $Ca^{2+}/Mg^{2+}$  etc.) preferentially combining with  $^{13}C$ -labeled  $C_2$  to form stable compounds like calcium oxalate. This mechanism is strongly supported by observational data during dust events, both oxalic acid and  $Ca^{2+}/Mg^{2+}$  concentrations showed increases with growing particle size, while exhibiting high correlations ( $R^2 = 0.30$ -0.65) during dust periods. Simultaneously,  $^{12}C$ -enriched  $C_2$  produced on fine particle surfaces moves to coarse particles through gas-particle conversion or coagulation, resulting in  $^{13}C$ -enriched residues remaining in fine particles. As demonstrated by the aforementioned research findings, the concentration of oxalic acid in coarse particles showed a marked increase during dust events, with this variation being particularly pronounced at the top. These two synergistic processes collectively altered both aerosol size distribution and isotopic composition characteristics.

# 4. Conclusions

Based on synchronous observations of PM<sub>2.5</sub> and size-segregated aerosols from surface to mountain sites, this study reveals the influence of dust transport on the formation pathways and  $\delta^{13}$ C signature of C<sub>2</sub> at different altitudes. The results indicate that during non-dust periods, C<sub>2</sub> is primarily generated through aqueous-phase oxidation in fine particles, whereas during dust periods, its formation shifts to heterogeneous chemical reactions on coarse-particle surfaces (Fig. 8).

At high-altitude sites, the influence of dust on aerosols was particularly pronounced, as evidenced by enhanced aerosol aging indicators ( $C_2/C_4$  ratio increased to 7.75), enriched  $\delta^{13}C$  (+8.3% compared to non-dust periods), and a distinct shift of  $C_2$  from the fine to the coarse mode. This redistribution resulted in a higher concentration of  $C_2$  in coarse particles (2301 ng m<sup>-3</sup>) than in fine particles (2161 ng m<sup>-3</sup>), producing a coarse-to-fine particle ratio of 1.1. The shift in particle size distribution was closely associated with the formation of  $C_4(NO_3)_2$  coatings during dust aging. These hygroscopic coatings provide active interfaces for the adsorption and oxidation of gaseous precursors, thereby promoting the formation of  $C_2$  and other SOA on coarse particle surfaces. The migration of  $NO_3$  from the fine mode (0.4–1.1 µm) to the coarse mode (3.3–5.8 µm) in the presence of  $Ca^{2+}$  provides direct evidence for the formation of such coatings. The pronounced enrichment of  $\delta^{13}C$  further supports this reaction pathway, which is

attributed to the synergistic effects of surface-catalyzed oxidation (favoring  $^{13}$ C retention) and metal-oxalate complexation. In contrast, aerosols at the low-altitude site were dominated by local anthropogenic emissions, exhibiting a lower degree of aging, as reflected by higher  $C_2$  concentrations (674  $\pm$  528 ng m<sup>-3</sup>) and lower  $\delta^{13}$ C values (-27.6‰). The higher background concentration of  $NO_3^-$  and competitive reactions with components such as  $SO_4^{2^-}$  collectively delayed the shift of  $NO_3^-$  to the coarse mode.

This study clarifies how dust alters the formation pathways, particle-size distribution, and  $\delta^{13}$ C signature of C<sub>2</sub>, highlighting the significant altitudinal variation of these effects. Our findings provide critical insights into the altitude-dependent transformation of SOA during dust transport, thereby enhancing the understanding of mountain atmospheric chemistry and regional climate effects. Limited by field observations, the microkinetic parameters of the aforementioned surface reactions remain unquantified. Future work should integrate laboratory experiments with model simulations to fully elucidate the microscopic mechanisms and regional climate impacts of this heterogeneous process.

# Data availability

The data in this study are available at <a href="https://zenodo.org/doi/10.5281/zenodo.15788834">https://zenodo.org/doi/10.5281/zenodo.15788834</a>
(Shen et al., 2025).

#### **Author contributions**

Jianjun Li conceived and designed the study. Minxia Shen conducted the literature search, performed sample and data analysis, and wrote the manuscript. Jianjun Li and Qiyuan Wang contributed to manuscript revision. Weining Qi, Yali Liu, Yifan Zhang, Wenting Dai, Lu Li, Xiao Guo, Yue Cao, Yingkun Jiang, Qian Wang and Shicong Li collected particulate samples and supervised the experiments. All authors provided critical feedback on the manuscript and approved the final version.

#### **Conflict of Interest**

The authors declare no conflicts of interest relevant to this study.

# Acknowledgments

This work was jointly supported by the program from National Natural Science

- Foundation of China (No. 42407156), State Key Laboratory of Loess and Quaternary Geology
- (SKLLOG2307), and the Natural Science Basic Research Program of Shaanxi Province
- (2025JC-YBQN-450). Jianjun Li also acknowledged the support of the Youth Innovation
- Promotion Association Chinese Academy of Sciences (No. 2020407).

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

Figure 1 Distribution of aerosol sampling sites at the foot and top of Mount Hua (December 2020 to January 2021)

Figure 2 Molecular distribution of dicarboxylic acids and related Compounds (a), relative percentages of major dicarboxylic acids (b), and ratios of  $C_2/C_4$  and  $C_3/C_4$  (c) at the foot and top of Mount Hua during non-dust periods

Figure 3 Correlation between C<sub>2</sub> and its key precursors, sulfate (SO<sub>4</sub><sup>2-</sup>), nitrate (NO<sub>3</sub><sup>-</sup>), and aerosol liquid water content (ALWC) at the foot and top of Mount Hua with varying relative humidity (RH)

Figure 4 Molecular distribution of dicarboxylic acids and related compounds (a), relative percentages of major dicarboxylic acids (b), and ratios of  $C_2/C_4$  and  $C_3/C_4$  (c) at the foot and top of Mount Hua during dust events

Figure 5 Size distribution of dicarboxylic acids at the foot and top of Mount Hua during non-dust and dust periods

Figure 6 Correlation of C<sub>2</sub> with water-soluble ions at the foot and top of Mount Hua during non-dust and dust periods (Circles in the figure represent non-dust periods, and triangles represent dust periods)

Figure 7 Stable carbon isotopes ( $\delta^{13}$ C) of  $C_2$  in  $PM_{2.5}$  at the foot and top of Mount Hua

Figure 8 Mechanism diagram of dust-driven particle-size migration and formation pathways of C2

|                                      |                | Foot          |                |               | Тор           |               |                  |  |  |  |
|--------------------------------------|----------------|---------------|----------------|---------------|---------------|---------------|------------------|--|--|--|
| Compound                             | Daytime        | Nighttime     | Whole          | Daytime       | Nighttime     | Whole         | R <sub>F/T</sub> |  |  |  |
|                                      | (N = 26)       | (N = 27)      | (N = 53)       | (N = 26)      | (N = 27)      | (N = 53)      |                  |  |  |  |
| Dicarboxylic acids                   |                |               |                |               |               |               |                  |  |  |  |
| Oxalic, C <sub>2</sub>               | $766 \pm 552$  | 585 ± 497     | $674 \pm 528$  | $312 \pm 224$ | 299 ± 186     | $306 \pm 204$ | 2.2              |  |  |  |
| Malonic, C <sub>3</sub>              | $86 \pm 58$    | $71 \pm 55$   | $78 \pm 56$    | $54 \pm 39$   | $53 \pm 32$   | $53 \pm 35$   | 1.5              |  |  |  |
| Succinic, C <sub>4</sub>             | $152 \pm 83$   | $112 \pm 77$  | $132 \pm 82$   | $60 \pm 47$   | $57 \pm 46$   | $58 \pm 46$   | 2.3<br>1.6       |  |  |  |
| Glutaric, C <sub>5</sub>             | $38 \pm 19$    | $27 \pm 14$   | $32 \pm 17$    | $22 \pm 27$   | $18 \pm 16$   | $20 \pm 22$   |                  |  |  |  |
| Adipic, C <sub>6</sub>               | $17 \pm 15$    | $14 \pm 13$   | $15 \pm 14$    | $12 \pm 13$   | $10 \pm 6.3$  | $11 \pm 9.8$  | 1.4              |  |  |  |
| Pimelic, C7                          | $15 \pm 5.0$   | $12 \pm 5.0$  | $13 \pm 5.1$   | $9.2 \pm 3.3$ | $8.9 \pm 2.3$ | $9.0\pm2.8$   | 1.4              |  |  |  |
| Suberic, C <sub>8</sub>              | $16 \pm 4.1$   | $14 \pm 3.4$  | $15\pm3.8$     | $13 \pm 3.7$  | $12\pm3.5$    | $12\pm3.6$    | 1.3              |  |  |  |
| Azelaic, C9                          | $175\pm110$    | $131\pm106$   | $153\pm110$    | $28\pm19$     | $27\pm17$     | $27\pm18$     | 5.7              |  |  |  |
| Sebacic, C <sub>10</sub>             | $19 \pm 15$    | $14 \pm 11$   | $16 \pm 14$    | $5.3 \pm 3.5$ | $5.3 \pm 3.7$ | $5.3 \pm 3.5$ | 3.0              |  |  |  |
| Undecanedioic, $C_{11}$              | $14\pm3.8$     | $12\pm2.5$    | $13 \pm 3.3$   | $9.9 \pm 1.8$ | $9.2 \pm 1.7$ | $9.6 \pm 1.8$ | 1.4              |  |  |  |
| $Methylmalonic, iC_4 \\$             | $36 \pm 17$    | $24 \pm 13$   | $29 \pm 16$    | $13 \pm 9.4$  | $12\pm7.6$    | $13 \pm 8.5$  | 2.2              |  |  |  |
| Mehtylsuccinic, iC5                  | $16 \pm 5.4$   | $16\pm7.9$    | $16\pm6.7$     | $10\pm3.2$    | $10\pm2.4$    | $10\pm2.8$    | 1.6              |  |  |  |
| Methylglutaric, iC <sub>6</sub>      | $18\pm6.7$     | $14 \pm 5.4$  | $16 \pm 6.3$   | $19\pm7.7$    | $19 \pm 6.2$  | $19\pm6.9$    | 0.8              |  |  |  |
| Maleic, M                            | $22 \pm 12$    | $17 \pm 11$   | $19 \pm 12$    | $13 \pm 10$   | $13 \pm 5.7$  | $13 \pm 7.9$  | 1.5              |  |  |  |
| Fumaric, F                           | $13 \pm 3.4$   | $11\pm2.4$    | $12\pm3.0$     | $10 \pm 1.2$  | $10 \pm 1.0$  | $9.7 \pm 1.1$ | 1.2              |  |  |  |
| Methylmaleic, mM                     | $29\pm18$      | $27\pm29$     | $28\pm24$      | $17 \pm 9.1$  | $16\pm6.2$    | $16\pm7.7$    | 1.8              |  |  |  |
| Phthalic, Ph                         | $89 \pm 43$    | $71 \pm 45$   | $80 \pm 44$    | $44 \pm 23$   | $39\pm20$     | $41\pm22$     | 2.0              |  |  |  |
| Isophthalic, iPh                     | $15\pm5.4$     | $13\pm7.0$    | $14\pm6.3$     | $6.5 \pm 1.2$ | $6.5 \pm 0.8$ | $6.5 \pm 1.0$ | 2.2              |  |  |  |
| Ketopimelic, kC7                     | $10.2 \pm 2.3$ | $9.4 \pm 2.1$ | $9.8 \pm 2.2$  | $8.3 \pm 1.9$ | $8.0 \pm 1.7$ | $8.1\pm1.8$   | 1.2              |  |  |  |
|                                      |                | Ket           | ocarboxylic ac | eids          |               |               |                  |  |  |  |
| Pyruvic, Pyr                         | 85 ± 45        | 73 ± 59       | 79 ± 53        | 52 ± 38       | 54 ± 30       | 53 ± 34       | 1.5              |  |  |  |
| Glyoxylic, $\omega C_2$              | $230\pm201$    | $198 \pm 200$ | $214 \pm 199$  | $79 \pm 58$   | $75 \pm 47$   | $77 \pm 52$   | 2.8              |  |  |  |
| α-Dicarbonyls                        |                |               |                |               |               |               |                  |  |  |  |
| Glyoxal, Gly                         | 29 ± 15        | 29 ± 23       | 29 ± 19        | 23 ± 13       | 24 ± 10       | 24 ± 12       | 1.2              |  |  |  |
| Methylglyoxal,<br>mGly               | 128 ± 49       | 87 ± 42       | 107 ± 49       | 49 ± 28       | $48\pm23$     | 48 ± 25       | 2.2              |  |  |  |
| Others                               |                |               |                |               |               |               |                  |  |  |  |
| Benzoic, Ha                          | 15 ± 7.1       | $12 \pm 6.7$  | 13 ± 6.9       | $9.6 \pm 5.4$ | $9.3 \pm 5.1$ | $9.4 \pm 5.2$ | 1.4              |  |  |  |
| Total detected (ng m <sup>-3</sup> ) | 2029 ±<br>1294 | 1594 ± 1238   | 1807 ± 1280    | 879 ± 591     | 842 ± 481     | $860 \pm 534$ | 2.1              |  |  |  |

|                                      |         | Foot      |               | Тор     |           |                                              |  |  |  |
|--------------------------------------|---------|-----------|---------------|---------|-----------|----------------------------------------------|--|--|--|
| Compound                             | Daytime | Nighttime | Whole         | Daytime | Nighttime | Whole                                        |  |  |  |
|                                      | (N = 1) | (N = 1)   | (N = 2)       | (N = 1) | (N=1)     | (N = 2)                                      |  |  |  |
| Oxalic, C <sub>2</sub>               | 289     | 262       | $276 \pm 20$  | 261     | 197       | $229 \pm 45$                                 |  |  |  |
| Malonic, C <sub>3</sub>              | 28      | 38        | $33 \pm 7.3$  | 21      | 21        | $21 \pm 0.1$<br>$30 \pm 3.5$<br>$10 \pm 1.9$ |  |  |  |
| Succinic, C <sub>4</sub>             | 47      | 59        | $53 \pm 8.9$  | 28      | 33        |                                              |  |  |  |
| Glutaric, C <sub>5</sub>             | 15      | 16        | $15 \pm 0.5$  | 8.4     | 11        |                                              |  |  |  |
| Adipic, C <sub>6</sub>               | 9.8     | 10.2      | $10\pm0.3$    | 6.2     | 6.4       | $6.3 \pm 0.2$                                |  |  |  |
| Pimelic, C <sub>7</sub>              | 7.6     | 9.2       | $8.4\pm1.1$   | 6.3     | 6.3       | $6.3 \pm 0.0$                                |  |  |  |
| Suberic, C <sub>8</sub>              | 9.2     | 11        | $10\pm1.3$    | 6.9     | 11        | $8.7 \pm 2.6$                                |  |  |  |
| Azelaic, C9                          | 41      | 54        | $47 \pm 9.4$  | 13      | 19        | $16 \pm 4.1$                                 |  |  |  |
| Sebacic, C <sub>10</sub>             | 4.7     | 5.6       | $5.1 \pm 0.7$ | 3.2     | 3.2       | $3.2 \pm 0.0$                                |  |  |  |
| Undecanedioic, $C_{11}$              | 9.3     | 10        | $9.8 \pm 0.7$ | 8.0     | 7.9       | $8.0 \pm 0.0$                                |  |  |  |
| Methylmalonic, iC <sub>4</sub>       | 15      | 16        | $16 \pm 0.3$  | 8.2     | 9.1       | $8.7 \pm 0.6$                                |  |  |  |
| Mehtylsuccinic, iC <sub>5</sub>      | 12      | 11        | $12\pm0.9$    | 9.4     | 10.2      | $9.8 \pm 0.6$                                |  |  |  |
| Methylglutaric, iC <sub>6</sub>      | 15      | 16        | $16 \pm 0.3$  | 9.5     | 10.0      | $9.8 \pm 0.3$                                |  |  |  |
| Maleic, M                            | 7.8     | 10        | $9.1\pm1.8$   | 7.3     | 5.6       | $6.4 \pm 1.3$                                |  |  |  |
| Fumaric, F                           | 13      | 15        | $14 \pm 1.4$  | 10.5    | 11.3      | $10.9 \pm 0.6$                               |  |  |  |
| Methylmaleic, mM                     | 12      | 13        | $13\pm0.2$    | 10.3    | 10.2      | $10.3\pm0.0$                                 |  |  |  |
| Phthalic, Ph                         | 39      | 41        | $40 \pm 1.3$  | 22      | 27        | $24 \pm 3.5$                                 |  |  |  |
| Isophthalic, iPh                     | 9.5     | 12        | $11 \pm 1.6$  | 5.9     | 5.9       | $5.9 \pm 0.0$                                |  |  |  |
| Ketopimelic, kC7                     | 7.5     | 7.9       | $7.7 \pm 0.3$ | 7.2     | 7.1       | $7.2 \pm 0.1$                                |  |  |  |
| Pyruvic, Pyr                         | 29      | 52        | $40 \pm 17$   | 0.6     | 19        | $9.8 \pm 13$                                 |  |  |  |
| Glyoxylic, ωC <sub>2</sub>           | 51      | 87        | $69\pm26$     | 33      | 38        | $36 \pm 4.0$                                 |  |  |  |
| Glyoxal, Gly                         | 14      | 8.7       | 12 ± 4.1      | 9.8     | 10.9      | 10.4± 0.7                                    |  |  |  |
| Methylglyoxal,<br>mGly               | 92      | 140       | 116 ± 34      | 32      | 44        | $38 \pm 8.6$                                 |  |  |  |
| Benzoic, Ha                          | 12      | 17        | $15 \pm 3.8$  | 8.7     | 10.7      | $9.7 \pm 1.4$                                |  |  |  |
| Total detected (ng m <sup>-3</sup> ) | 761     | 949       | 855 ± 142     | 535     | 533       | $534 \pm 92$                                 |  |  |  |

Table 3 Comparison of concentrations of dicarboxylic acids and related compounds in particulate matter of different particle size ranges ( $\leq$ 2.1  $\mu$ m and >2.1  $\mu$ m) at the foot and top of Mount Hua during non-dust and dust periods

|                                        | Foot        |            |                      |            |            | Тор        |            |            |                  |            |            |            |
|----------------------------------------|-------------|------------|----------------------|------------|------------|------------|------------|------------|------------------|------------|------------|------------|
| Commound                               | Non-dust Du |            | ıst R <sub>D/N</sub> |            | Non-dust   |            | Dust       |            | R <sub>D/N</sub> |            |            |            |
| Compound                               | ≤2.1<br>μm  | >2.1<br>μm | ≤2.1<br>μm           | >2.1<br>μm | ≤2.1<br>μm | >2.1<br>μm | ≤2.1<br>μm | >2.1<br>µm | ≤2.1<br>μm       | >2.1<br>µm | ≤2.1<br>μm | >2.1<br>μm |
| Oxalic, C <sub>2</sub>                 | 8662        | 2718       | 4880                 | 2843       | 0.3        | 0.6        | 5512       | 2217       | 2161             | 2301       | 0.4        | 1.1        |
| Malonic, C <sub>3</sub>                | 750         | 388        | 361                  | 371        | 0.5        | 1.0        | 757        | 359        | 305              | 253        | 0.5        | 0.8        |
| Succinic, C <sub>4</sub>               | 1505        | 594        | 829                  | 505        | 0.4        | 0.6        | 986        | 488        | 371              | 294        | 0.5        | 0.8        |
| Glutaric, C <sub>5</sub>               | 238         | 350        | 199                  | 297        | 1.5        | 1.5        | 557        | 390        | 334              | 324        | 0.7        | 1.0        |
| Adipic, C <sub>6</sub>                 | 427         | 277        | 347                  | 292        | 0.6        | 0.8        | 457        | 275        | 290              | 240        | 0.6        | 0.8        |
| Azelaic, C9                            | 2313        | 371        | 1097                 | 304        | 0.2        | 0.3        | 646        | 302        | 379              | 396        | 0.5        | 1.0        |
| Methylglyoxal,<br>mGly                 | 237         | 119        | 217                  | 90         | 0.5        | 0.4        | 146        | 67         | 49               | 55         | 0.5        | 1.1        |
| Glyoxal, Gly                           | 312         | 268        | 237                  | 145        | 0.9        | 0.6        | 192        | 129        | 140              | 114        | 0.7        | 0.8        |
| Pyruvic, Pyr                           | 729         | 800        | 687                  | 513        | 1.1        | 0.7        | 1022       | 521        | 451              | 633        | 0.5        | 1.4        |
| Glyoxylic, ωC <sub>2</sub>             | 2181        | 1171       | 979                  | 663        | 0.5        | 0.7        | 1138       | 678        | 658              | 545        | 0.6        | 0.8        |
| Phthalic, Ph                           | 1018        | 652        | 499                  | 386        | 0.6        | 0.8        | 564        | 481        | 302              | 356        | 0.9        | 1.2        |
| Isophthalic, iPh (ng m <sup>-3</sup> ) | 206         | 184        | 157                  | 163        | 0.9        | 1.0        | 162        | 173        | 146              | 169        | 1.1        | 1.2        |