# Peer review of "(untitled)"

_EGUsphere, 2025_

## Referee Comment (RC1)

**General comments:**

This study conducted simultaneous observations at the foot and top of Mount Hua, systematically analyzing the distribution characteristics of dicarboxylic acids (with a focus on oxalic acid, $C_2$) in $PM_{2.5}$ and size-segregated samples, along with their $\delta^{13}C$ composition during both non-dust and dust periods. The research data demonstrate novelty and provide important evidence for understanding the transformation processes of SOA and the interactions between dust and anthropogenic emissions in mountainous environments. While the manuscript is fundamentally sound, certain aspects require further clarification and refinement.

1. The introduction would benefit from smoother transitions between paragraphs. Currently, there's a noticeable jump from discussing research gaps (end of paragraph 2) to introducing nitrate-aged dust effects (paragraph 3) without adequate transitional phrasing.

2. Line 64-66: "Resultsshow that dust storms have a less pronounced impact on ground aerosols than on the free troposphere of the Guanzhong Plain (Liu et al., 2024)..." Please specify the exact chemical species (e.g., water-soluble ions, carbonaceous fractions) and the particle-size ranges examined by Liu et al., 2024, and clarify whether their sampling sites coincide with those of the present study to avoid any potential confusion.

3. Further refinement of language and optimization of sentence structure are needed. For instance, line 46-47, "Dust degrades air quality near its source and can be transported over long distances by winds, impacting the climate on hemispheric and global scales" could be improved to: "Dust not only impairs air quality locally but also undergoes long-range transport, ultimately affecting both hemispheric and global climate systems"; Line 60-62: "Originating in southern Mongolia and China's western Inner Mongolia, it was intense and far-reaching, causing rapid air quality deterioration

in the affected areas" can be revised as "This severe dust storm, originating from southern Mongolia and western Inner Mongolia, triggered rapid air quality deterioration across downwind regions".

4. Line 62: The introduction merely cites Figure S1 without clarifying its relevance. Revise the sentence so that specific elements of the figure are explicitly linked to the research questions.

5. Maintain terminological consistency: "secondary organic aerosols" was abbreviated as "SOA" in line 68; please use "SOA" throughout the manuscript (including lines 102 and 360) and avoid alternating with the full term.

6. Unify citation format: for example, line 38 "(Maher et al. 2010; Liang et al., 2022)" should be corrected to "(Maher et al., 2010; Liang et al., 2022)" with the comma added; please check the entire manuscript.

7. Line 121-139: The analytical methods lack sufficient detail. Please provide the exact column model, detection limits for target ions (e.g., $Cl^-$, $NO_3^-$, $SO_4^{2-}$) and for OC/EC, together with a full description of the drying protocol for dicarboxylic acid samples, the hexane-wash purification of their organic derivatives, and the nitrogen blow-down concentration procedure.

8. Line 129: Please specify the exact source(s) of the relative humidity and temperature data.

9. The discussion of results does not adequately compare with key reference. For example, although the study points out that oxalic acid ($C_2$) at the foot shows significant diurnal variations, while the top shows no significant differences, it does not compare these findings with reference on photochemical processes in similar topographical studies. Moreover, the study uses the $C_2/C_4$ ratio of 5.84 (at the top) as

an indicator of photochemical aging but fails to cite the classic study by Kawamura and Ikushima (1993), who first proposed that this ratio can indicate the degree of aging. It is recommended to supplement the study with references to Kawamura and Ikushima (1993) and other relevant studies in high-altitude regions to further substantiate the applicability of the $C_2/C_4$ ratio as an aging indicator.

10. Please provide an explanation for the notable discrepancy observed between the nitrate-dominated $C_2$ correlation (with an $R^2$ value of 0.79) that you have observed and the previously reported sulfate-dominated mechanisms by Meng et al. (2018) and Wang et al. (2012).

11. Line 265-268: the authors mention "the proportion of $C_2$ decreasing from 37.3% to 32.2% at the foot and increasing from 35.5% to 42.8% at the top" Is it sufficient to explain this solely based on humidity? Additionally, they mention that high humidity promotes the "secondary formation of $C_2$" Is this also referring to an aqueous-phase reaction?

12. It is recommended to divide the 16 subplots of Figure 3 into two groups for discussion: the correlations between precursors and $C_2$ at the foot and summit sites (Figures 3a-d, i-l), and the influences of inorganic ions and ALWC on $C_2$ (Figures 3e-h, m-p).

13. Please explicitly cite the relevant figure numbers in the text, for example by adding "(Figure 4c)" after the mention of "$C_2/C_4$ ratios" on line 282, to strengthen the linkage between the narrative and the figures.

14. Line 340-341: the authors need to clarify whether the conclusion that "aqueous secondary organic aerosols (aqSOA) formed on dust surfaces promote SOA formation and drive the transition of particle size distribution from submicron to supermicron

ranges" is based on the findings of Li et al. (2025) or represents original findings from this study. If it is a reference to the literature, it is recommended to clearly indicate the citation.

15. Line 364-366: Please ensure consistency in verb tense, and it is recommended to replace "oxalate" with "$C_2$" for consistency in terminology usage.

16. Please add the p-values for all correlation coefficients in Figures 3 and 6 so that readers can properly assess their statistical significance. Additionally, standardize the notation for dicarboxylic acids in Table 3 to match that used in Tables 1 and 2.

17. Line 442: "coarse/fine ratio" should be revised to "coarse/fine particle ratio" for greater precision.

---

## Author Comment (AC2)

**Reviewer comments:**

**Reviewer #1:**

**General comment:**

This study conducted simultaneous observations at the foot and top of Mount Hua, systematically analyzing the distribution characteristics of dicarboxylic acids (with a focus on oxalic acid,  $C_2$ ) in  $PM_{2.5}$  and size-segregated samples, along with their  $\delta^{13}C$  composition during both non-dust and dust periods. The research data demonstrate novelty and provide important evidence for understanding the transformation processes of SOA and the interactions between dust and anthropogenic emissions in mountainous environments. While the manuscript is fundamentally sound, certain aspects require further clarification and refinement.

**Response:** We sincerely appreciate the reviewer's positive assessment and constructive comments, which are very helpful for improving our manuscript. Our point-by-point responses and corresponding revisions are detailed below.

1. The introduction would benefit from smoother transitions between paragraphs. Currently, there's a noticeable jump from discussing research gaps (end of paragraph 2) to introducing nitrate-aged dust effects (paragraph 3) without adequate transitional phrasing.

**Response:** We thank the reviewer for this constructive suggestion. We have improved the logical flow by adding a transitional sentence at the beginning of the third paragraph. This sentence connects the research gap mentioned at the end of the second paragraph (unclear mechanisms of dust-organic interactions) to the specific focus of our study (dicarboxylic acids as key tracers). The revised text reads as follows:

To investigate these processes, this study focuses on dicarboxylic acids (diacids), which serve as key tracers for SOA (Xu et al., 2022). As important components of water-soluble organic carbon, diacids are widely distributed in the atmosphere from the surface layer to the free troposphere (Fu et al., 2008; Myriokefalitakis et al., 2011).

2. Line 64-66: "Results show that dust storms have a less pronounced impact on ground aerosols than on the free troposphere of the Guanzhong Plain (Liu et al., 2024) ..." Please specify the exact chemical species (e.g., water-soluble ions, carbonaceous fractions) and the particle- size ranges examined by Liu et al., 2024, and clarify whether their sampling sites coincide with those of the present study to avoid any potential confusion.

**Response:** We thank the reviewer for this suggestion. We have clarified that Liu et al. (2024) was conducted concurrently at our study sites, analyzing water-soluble inorganic ions in  $PM_{2.5}$  and size-segregated aerosols. The relevant sentence now reads:

Liu et al. (2024) compared and analyzed the concentrations and size distributions of water-soluble inorganic ions during dust and non-dust periods, finding that the impact of dust on ground aerosols in the Guanzhong Plain is weaker than that in the free troposphere.

3. Further refinement of language and optimization of sentence structure are needed. For instance, line 46-47, "Dust degrades air quality near its source and can be transported over

long distances by winds, impacting the climate on hemispheric and global scales" could be improved to: "Dust not only impairs air quality locally but also undergoes long-range transport, ultimately affecting both hemispheric and global climate systems"; Line 60-62: "Originating in southern Mongolia and China's western Inner Mongolia, it was intense and far-reaching, causing rapid air quality deterioration in the affected areas" can be revised as "This severe dust storm, originating from southern Mongolia and western Inner Mongolia, triggered rapid air quality deterioration across downwind regions".

**Response:** We thank the reviewer for these constructive suggestions to improve the clarity and conciseness of our manuscript. Following the reviewer's specific recommendations, we have revised the two sentences accordingly:

The sentence in Lines 46-47 has been revised to: "Dust not only impairs air quality locally but also undergoes long-range transport, ultimately affecting both hemispheric and global climate systems."

The sentence in Lines 60-62 has been revised to: "This severe dust storm, originating from southern Mongolia and western Inner Mongolia, triggered rapid air quality deterioration across downwind regions."

In addition, we have performed a comprehensive language check throughout the manuscript to refine phrasing and optimize clarity.

4. Line 62: The introduction merely cites Figure S1 without clarifying its relevance. Revise the sentence so that specific elements of the figure are explicitly linked to the research questions.

**Response:** We thank the reviewer for this suggestion. We have revised the sentence to explicitly link Figure S1 to the spatial extent of the dust event. The updated text now reads:

Our synchronized field observations of PM2.5 and size-segregated aerosols at the top of Mount Hua and on the ground in the winter of 2021 successfully captured this large-scale dust event, as shown in Fig. S1, extensively covering Northern China and the Guanzhong Plain.

5. Maintain terminological consistency: "secondary organic aerosols" was abbreviated as "SOA" in line 68; please use "SOA" throughout the manuscript (including lines 102 and 360) and avoid alternating with the full term.

**Response:** We thank the reviewer for pointing this out. The abbreviation "SOA" has been used consistently throughout the revised manuscript, including in Lines 102 and 360.

6. Unify citation format: for example, line 38 "(Maher et al. 2010; Liang et al., 2022)" should be corrected to "(Maher et al., 2010; Liang et al., 2022)" with the comma added; please check the entire manuscript.

**Response:** We appreciate the reviewer's careful attention to detail. We have corrected the citation format (e.g., in Line 38) and have thoroughly checked the entire manuscript to ensure that all in-text citations now follow the consistent format of "(Author et al., Year)".

7. Line 121-139: The analytical methods lack sufficient detail. Please provide the exact column model, detection limits for target ions (e.g., Cl-, NO3-, SO42-) and for OC/EC, together with a full description of the drying protocol for dicarboxylic acid samples, the

hexane-wash purification of their organic derivatives, and the nitrogen blow-down concentration procedure.

**Response:** We sincerely thank the reviewer for this constructive suggestion. We have now expanded the Methods section to provide comprehensive details regarding the analytical procedures, as requested. The modifications are presented in the revised manuscript (Section 2.2.1 and 2.2.2) and summarized below:

**2.2.1 Determination of carbonaceous species and water-soluble inorganic ions**

The concentrations of organic carbon (OC) and elemental carbon (EC) in PM2.5 were determined using a DRI Model 2001 carbon analyzer (Atmoslytic Inc., USA), following the IMPROVE thermal/optical reflectance (TOR) protocol (Cao et al., 2007). A 0.526 cm2 filter punch was heated stepwise in pure helium (at 120 °C, 250 °C, 450 °C, and 550 °C) followed by heating in a 2% oxygen/helium atmosphere (at 550 °C, 700 °C, 800 °C). The method detection limits were 0.41 μg cm-2 for OC and 0.03 μg cm-2 for EC.

Water-soluble components were extracted from a quarter of each filter using 40 mL of ultrapure water (Milli-Q, 18.2 M $\Omega$ , Merck, France) via a combined process of 1-hour ultrasonication and 1-hour mechanical shaking. After filtration through a 0.45  $\mu$ m membrane, the extracts were preserved at 4 °C for subsequent analysis. Water-soluble inorganic ions (NH4+, K+, Mg2+, Ca2+, Cl-, NO3-, and SO42-) were analyzed by ion chromatography (Metrohm 940, Switzerland). Anions and cations were separated using an IonPac AS23 column and an IonPac CS12A column, with 9.0 mM Na2CO3 and 20 mM methanesulfonic acid as eluents, respectively (Zhang et al., 2011). Concurrently, water-soluble organic carbon (WSOC) was determined using a total organic carbon (TOC) analyzer (TOC-L CPH, Shimadzu, Japan) (Li et al., 2019). The detection limits for inorganic ions ranged from 0.008 to 0.022  $\mu$ g m-3, while those for total carbon (TC) and inorganic carbon (IC) were 0.07 mg L-1 and 0.08 mg L-1, respectively.

Aerosol liquid water content (ALWC) was calculated using the ISORROPIA-II model (Fountoukis and Nenes, 2007), based on the concentrations of water-soluble inorganic ions and meteorological parameters including relative humidity (RH) and temperature (T). Meteorological data for the top and the foot were obtained from the Mount Hua Meteorological Station and the Huayin Meteorological Bureau, respectively. All statistical analyses were performed using SPSS.

**2.2.2 Determination of dicarboxylic acids and related compounds**

The analysis of diacids, keto-carboxylic acids, and α-dicarbonyls in PM2.5 and size-segregated aerosols was conducted based on an established derivatization method (Shen et al., 2022). A quarter of each filter was extracted by ultrasonicating in ultrapure water (Milli-Q, 18.2 MΩ, Merck, France) for three sequential 15-minute intervals. To maximize the recovery of low-molecular-weight acids (e.g., C2), the extracts were alkalized to pH 8.5–9.0 with 0.1 M KOH prior to the concentration step. The aqueous extracts were concentrated to near-dryness under vacuum in a water bath maintained at 55 °C (evaporation was halted immediately after the disappearance of the last drop of solvent). The dried residues were derivatized with 14% BF3/n-butanol at 100°C for 1 hour to convert carboxyl groups into dibutyl esters and oxo groups into dibutoxyacetals. After the reaction, the derivatives were sequentially dissolved in n-hexane, acetonitrile, and pure water, followed by triple extraction

via vortex mixing to remove water-soluble inorganic substances. The aqueous lower layer was removed using a Pasteur pipette. The n-hexane layer was concentrated by rotary evaporation and  $N_2$  blow-down, reconstituted in 100  $\mu$ L n-hexane, and finally analyzed by GC-FID (HP 6890, Agilent Technologies, USA) (Wang et al., 2012). The recovery of the target compounds was 83% for  $C_2$  and ranged from 87% to 110% for other diacids.

Stable carbon isotopic compositions ( $\delta^{13}$ C) of  $C_2$  were determined using gas chromatography–isotope ratio mass spectrometry (GC-IRMS; Delta V Advantage, Thermo Fisher Scientific, Franklin, MA, USA) following established protocols (Kawamura and Watanabe, 2004). To ensure analytical precision (standard deviation < 0.2‰), derivatized samples were analyzed in triplicate. Final  $\delta^{13}$ C values of free oxalic acid were calculated via mass-balance correction to account for isotopic contributions from the BF3/n-butanol derivatizing agent.

8. Line 129: Please specify the exact source(s) of the relative humidity and temperature data.

**Response:** We thank the reviewer for this comment. The requested clarification has been added to the manuscript. The revised text now reads: "Meteorological data for the top and the foot were obtained from the Mount Hua Meteorological Station and the Huayin Meteorological Bureau, respectively."

9. The discussion of results does not adequately compare with key reference. For example, although the study points out that oxalic acid ( $C_2$ ) at the foot shows significant diurnal variations, while the top shows no significant differences, it does not compare these findings with reference on photochemical processes in similar topographical studies. Moreover, the study uses the  $C_2/C_4$  ratio of 5.84 (at the top) as an indicator of photochemical aging but fails to cite the classic study by Kawamura and Ikushima (1993), who first proposed that this ratio can indicate the degree of aging. It is recommended to supplement the study with references to Kawamura and Ikushima (1993) and other relevant studies in high-altitude regions to further substantiate the applicability of the  $C_2/C_4$  ratio as an aging indicator.

**Response:** We fully agree with the reviewer's comment. Following the suggestion, we have now cited the study by Kawamura and Ikushima (1993) when discussing the  $C_2/C_4$  and  $C_3/C_4$  ratios as aging indicators. Additionally, we have incorporated comparative analyses with relevant studies from high-altitude areas such as Mount Hua and Mount Tai, which further validate the reliability of our findings. The revised text now reads:

Aerosol aging indicators further confirmed distinct oxidation processes between different altitudes. According to existing research, succinic acid ( $C_4$ ) can be via hydroxylation to generate  $C_2$  and malonic acid ( $C_3$ ), while  $C_3$  can be further converted to  $C_2$  through intermediates such as hydroxymalonic acid or ketomalonic acid (Kawamura et al. 1993; Kunwar and Kawamura, 2014; Hoque et al., 2017). Therefore, the  $C_2/C_4$  and  $C_3/C_4$  ratios are widely used as effective indicators for assessing the extent of photochemical aging in organic aerosols (Kawamura et al., 2016; Meng et al., 2018; Shen et al., 2022). In this study, the  $C_2/C_4$  ratio at the top of Mount Hua was  $5.84 \pm 0.32$ , and the  $C_3/C_4$  ratio was  $1.04 \pm 0.08$ , both significantly higher than the corresponding values at the foot ( $4.74 \pm 0.28$  and  $0.56 \pm 0.05$ , respectively; Fig. 2c). These results are consistent with reports from high-altitude areas such

as Mount Tai (Wang et al., 2009; Meng et al., 2018) and Mount Hua (Meng et al., 2014), collectively confirming that the atmosphere undergoes more significant photochemical aging during long-range transport at high altitudes.

10. Please provide an explanation for the notable discrepancy observed between the nitrate-dominated  $C_2$  correlation (with an  $R^2$  value of 0.79) that you have observed and the previously reported sulfate-dominated mechanisms by Meng et al. (2018) and Wang et al. (2012).

**Response:** Thank you for your valuable suggestion. We have added an explanation for this discrepancy in the discussion section of the manuscript, as detailed below:

Contrary to previous studies (Wang et al., 2012; Meng et al., 2018), we found that  $C_2$  correlated more strongly with  $NO_3^-$  ( $R^2 = 0.79$ , p < 0.01; Fig. 3f) than with  $SO_4^{2-}$  ( $R^2 = 0.71$ , p < 0.01; Fig. 3e) at the foot. The difference mainly stems from strong influences of local anthropogenic emissions (e.g., traffic and industrial activities) at the foot, which provide abundant  $NO_x$  and lead to an increased proportion of  $NO_3^-$  in  $PM_{2.5}$ . Since  $NO_3^-$  is more hygroscopic than  $SO_4^{2-}$ , its elevated concentration further enhances ALWC, thereby promoting aqSOA formation and intensifying heterogeneous reaction processes (Huang et al., 2025). In contrast, at the top of Mount Hua,  $C_2$  exhibited a higher correlation with  $SO_4^{2-}$  ( $R^2 = 0.63$ , p < 0.01; Fig. 3m) than with  $NO_3^-$  ( $R^2 = 0.37$ , p < 0.01; Fig. 3n). This phenomenon is closely related to the active in-cloud processes in the free troposphere mentioned above.  $SO_4^{2-}$  at the top primarily originates from the oxidation of  $SO_2$  within cloud droplets (Yermakov et al., 2023), a process dominated by heterogeneous reactions (Wang et al., 2025). Meanwhile, cloud droplets also provide key reaction media for the aqueous phase photooxidation of  $C_2$  precursors such as Gly and  $\omega C_2$  (Warneck, 2003).

11. Line 265-268: the authors mention "the proportion of  $C_2$  decreasing from 37.3% to 32.2% at the foot and increasing from 35.5% to 42.8% at the top" Is it sufficient to explain this solely based on humidity? Additionally, they mention that high humidity promotes the "secondary formation of  $C_2$ " Is this also referring to an aqueous-phase reaction?

**Response:** We thank the reviewer for this important comment. We agree that explaining the trend based solely on humidity is insufficient, and we confirm that the "secondary formation of C2" in our context primarily refers to aqueous-phase reactions. We have revised the manuscript accordingly to provide a more comprehensive explanation.

During the dust event, the lower relative humidity at the foot (RH =  $24 \pm 8.5\%$ ) suppressed aqueous-phase oxidation, whereas the higher humidity at the top (RH =  $44 \pm 11\%$ ) favored C2 formation through such reactions. Additionally, variations in aerosol sources, transport pathways, aging processes, and potential contributions from other chemical reactions may also have influenced C2 generation.

12. It is recommended to divide the 16 subplots of Figure 3 into two groups for discussion: the correlations between precursors and C2 at the foot and summit sites (Figures 3a-d, i-l), and the influences of inorganic ions and ALWC on C2 (Figures 3e-h, m-p).

**Response:** We thank the reviewer for this constructive suggestion. The discussion of Fig. 3

has been reorganized into two distinct sections as recommended the correlations between C2 and its key precursors at the foot and summit sites (Fig. 3a–d, i–l), and the influences of inorganic ions and ALWC on C2 formation (Fig. 3e–h, m–p). The corresponding text has been revised to reflect this clearer structure and now reads:

This study compared the correlations between C2 and its key precursors at the foot and top of Mount Hua (Fig. 3a–d, 3i–l), revealing differences in atmospheric oxidation pathways across altitudes. Building on this, the synergistic effects of inorganic ions and ALWC on C2 formation were investigated (Fig. 3e–h, 3m–p), thereby elucidating the aqueous-phase reaction mechanisms and the role of environmental factors such as humidity in C2 generation.

13. Please explicitly cite the relevant figure numbers in the text, for example by adding "(Figure 4c)" after the mention of " $C_2/C_4$  ratios" on line 282, to strengthen the linkage between the narrative and the figures.

**Response:** We thank the reviewer for this suggestion. We have now added explicit citations to the relevant figure panels throughout the manuscript to strengthen the link between the narrative and the data. For example, as suggested, we have added "(Fig. 4c)" following the mention of the " $C_2/C_4$  ratios" on line 282. A thorough check has been performed to ensure all relevant figures are properly cited.

14. Line 340-341: the authors need to clarify whether the conclusion that "aqueous secondary organic aerosols (aqSOA) formed on dust surfaces promote SOA formation and drive the transition of particle size distribution from submicron to supermicron ranges" is based on the findings of Li et al. (2025) or represents original findings from this study. If it is a reference to the literature, it is recommended to clearly indicate the citation.

**Response:** We thank the reviewer for this important clarification. We have revised the sentence in question by explicitly citing Li et al. (2025) to attribute the conclusion that aqSOA formed on dust surfaces promotes SOA production and drives the size-distribution transition. The original general statement has been replaced with the following text:

Research by Li et al. (2025) further confirms that aqSOA formed on dust surfaces can effectively enhance SOA production and drive a transition in the size distribution from the submicron to the supermicron range, which is highly consistent with the observational results of this study.

15. Line 364-366: Please ensure consistency in verb tense, and it is recommended to replace "oxalate" with "C2" for consistency in terminology usage.

**Response:** We thank the reviewer for this careful observation. We have corrected the verb tense inconsistency in Lines 364–366. Regarding terminology, we have conducted a thorough review and now use "C2" as the primary term throughout the manuscript to ensure consistency. The term "oxalate" is retained exclusively in contexts that specifically describe its salt form with metal cations (e.g., calcium oxalate) to maintain chemical accuracy.

16. Please add the p-values for all correlation coefficients in Figures 3 and 6 so that readers can properly assess their statistical significance. Additionally, standardize the notation for dicarboxylic acids in Table 3 to match that used in Tables 1 and 2.

**Response:** We thank the reviewer for these valuable suggestions. We have added the p-values for all correlation coefficients in Fig. 3 and 6 to allow for a proper assessment of their statistical significance and have standardized the notation for dicarboxylic acids in Table 3 to ensure consistency with the format used in Tables 1 and 2.

Figure 3 Correlation between C2 and its key precursors, sulfate (SO42-), nitrate (NO3-), and aerosol liquid water content (ALWC) at the foot and top of Mount Hua with varying relative humidity (RH)

Figure 6 Correlation of C2 with water-soluble ions at the foot and top of Mount Hua during non-dust and dust periods (Circles in the figure represent non-dust periods, and triangles represent dust periods)

Table 3 Comparison of concentrations of dicarboxylic acids and related compounds in particulate matter of different particle size ranges ( $\leq$ 2.1  $\mu$ m and >2.1  $\mu$ m) at the foot and top of Mount Hua during non-dust and dust periods

|                                        | Foot       |            |            |            | Тор        |            |            |            |            |            |            |            |
|----------------------------------------|------------|------------|------------|------------|------------|------------|------------|------------|------------|------------|------------|------------|
| Compound                               | Non        | -dust      | Dı         | ust        | R          | D/N        | Non        | -dust      | D          | ust        | R          | D/N        |
|                                        | ≤2.1
μm | >2.1
μm |
| Oxalic, C 2                 | 8662       | 2718       | 4880       | 2843       | 0.3        | 0.6        | 5512       | 2217       | 2161       | 2301       | 0.4        | 1.1        |
| Malonic, C 3                | 750        | 388        | 361        | 371        | 0.5        | 1.0        | 757        | 359        | 305        | 253        | 0.5        | 0.8        |
| Succinic, C 4               | 1505       | 594        | 829        | 505        | 0.4        | 0.6        | 986        | 488        | 371        | 294        | 0.5        | 0.8        |
| Glutaric, C 5               | 238        | 350        | 199        | 297        | 1.5        | 1.5        | 557        | 390        | 334        | 324        | 0.7        | 1.0        |
| Adipic, C 6                 | 427        | 277        | 347        | 292        | 0.6        | 0.8        | 457        | 275        | 290        | 240        | 0.6        | 0.8        |
| Azelaic, C9                            | 2313       | 371        | 1097       | 304        | 0.2        | 0.3        | 646        | 302        | 379        | 396        | 0.5        | 1.0        |
| Methylglyoxal,
mGly                 | 237        | 119        | 217        | 90         | 0.5        | 0.4        | 146        | 67         | 49         | 55         | 0.5        | 1.1        |
| Glyoxal, Gly                           | 312        | 268        | 237        | 145        | 0.9        | 0.6        | 192        | 129        | 140        | 114        | 0.7        | 0.8        |
| Pyruvic, Pyr                           | 729        | 800        | 687        | 513        | 1.1        | 0.7        | 1022       | 521        | 451        | 633        | 0.5        | 1.4        |
| Glyoxylic, ωC 2             | 2181       | 1171       | 979        | 663        | 0.5        | 0.7        | 1138       | 678        | 658        | 545        | 0.6        | 0.8        |
| Phthalic, Ph                           | 1018       | 652        | 499        | 386        | 0.6        | 0.8        | 564        | 481        | 302        | 356        | 0.9        | 1.2        |
| Isophthalic, iPh (ng m -3 ) | 206        | 184        | 157        | 163        | 0.9        | 1.0        | 162        | 173        | 146        | 169        | 1.1        | 1.2        |

17. Line 442: "coarse/fine ratio" should be revised to "coarse/fine particle ratio" for greater precision.

**Response:** We thank the reviewer for this suggestion to improve terminological precision. We have revised "coarse/fine ratio" to "coarse/fine particle ratio" in Line 442 as recommended.


Response: We sincerely thank the reviewer for this insightful comment and for highlighting the importance of explicitly citing the role of aqueous nitrate coatings in facilitating SOA formation. We fully agree that the mechanism of Ca(NO3)2 induced aqueous-phase reactions under low-RH conditions is critical to our discussion. We have revised the manuscript accordingly to incorporate the suggested reference and to more clearly articulate the deliquescence behavior and its implications for SOA formation. The modified text now reads as follows:

To investigate these processes, this study focuses on dicarboxylic acids (diacids), which serve as key tracers for SOA (Xu et al., 2022). As important components of water-soluble organic carbon, diacids are widely distributed in the atmosphere from the surface layer to the free troposphere (Fu et al., 2008; Myriokefalitakis et al., 2011). Conventional theory suggests that aqueous-phase chemical reactions occur predominantly in submicron particles containing water or cloud droplets (Lim et al., 2010; Ervens et al., 2011; Lamkaddam et al., 2021). However, field observations have reported the coexistence of oxalate and nitrate in supermicron particles during dust events (Falkovich and Schkolnik, 2004; Sullivan et al., 2007a; Wang et al., 2015; Xu et al., 2020). To explain this, Wang et al. (2015) proposed that the reaction of nitric acid and/or nitrogen oxides with dust generates (Ca(NO3)2), which absorbs water vapor to form an aqueous phase on the dust surface. This enables the partitioning of gas-phase water-soluble organic precursors into this aqueous phase, followed by their further oxidation to form oxalic acid (C2). Research by Li et al. (2025) provides direct evidence for this mechanism, showing that aqueous nitrate coatings (Ca(NO3)2), due to their very low deliquescence relative humidity (absorbing water at atmospheric RH > 8%),

effectively promote the formation of aqueous secondary organic aerosols (aqSOA). Thus, aged dust surfaces provide critical reactive interfaces for aqSOA formation.

2. Statistical support for "most striking differences" throughout your manuscript. Wherever you describe significant differences (e.g., foot vs. top; dust vs. non-dust; day vs. night), please perform and report statistical tests (e.g., two-sample t-tests with assumptions stated). The manuscript currently lacks the statistical support needed to substantiate claims of significance.

Response: Thank you very much for your valuable comments. In response to your suggestions, we have now supplemented the manuscript with complete statistical analyses for all comparative conclusions to provide solid statistical support. Specifically, independent samples t-tests were used for the comparison between the foot and top of the mountain during non-dust periods, with detailed results provided in Supplementary Tables S3a and S3b; paired samples t-tests were used for the day-night variation comparisons, with detailed results shown in Tables S4a, S4b, and S4c. All descriptions such as "most striking" or "significantly higher" now explicitly cite the specific statistical results (including t-value, degrees of freedom, p-value, and 95% confidence interval). Furthermore, for the dust events with a limited sample, statistical testing was not performed, and the relevant descriptions have been adjusted accordingly.

During non-dust periods, significant vertical differences were observed in the chemical composition of PM2.5 between the foot and top of Mount Hua. The average PM2.5 concentration at the foot (127  $\pm$  48  $\mu$ g m-3) was 4.5 times higher than that at the top (28  $\pm$  14  $\mu$ g m-3), with a mean difference of 99  $\mu$ g m-3 (95% CI: 86 to 113; t (61.05) = 14.60, p < 0.001) (Supplementary Table S1 and Table S3a). Carbonaceous aerosols and water-soluble ions were significantly enriched at the foot (all p < 0.001; Table S3a). As key components of WSOC (Kawamura et al., 2016; Yang et al., 2020), the total concentration of diacids at the foot (1808  $\pm$  1280 ng m-3) was approximately 2.1 times higher than those at the top (860  $\pm$  534 ng m-3), with a mean difference of 718 ng m-3 (95% CI: 453 to 983; t (69.15) = 5.40, p < 0.001; Table S3b). The spatial difference was most pronounced for azelaic acid (C9), a biomarker of biomass burning (Kalogridis et al., 2018; Shen et al., 2022), with the concentration at the foot (153  $\pm$  110 ng m-3) being 5.7 times higher than that at the top (27  $\pm$  18 ng m-3), corresponding to a mean difference of 126 ng m-3 (95% CI: 95 to 156; t (54.77) = 8.24, p < 0.001; Table S3b). Consequently, the contribution of C9 to total diacids was substantially greater at the foot (8.5%) than at the top (3.2%) (Fig. 2b).

At the foot of the mountain,  $PM_{2.5}$ , ionic components, and carbonaceous components (except for  $NO_3^-$  and  $NH_4^+$ ) all showed significant day-night differences (p < 0.05; Table S4a). Among the diacids, methylglyoxal (mGly) exhibited the most pronounced day-night differences (daytime:  $128 \pm 49$  ng m-3, 47% higher than nighttime:  $87 \pm 42$  ng m-3, p < 0.001; Fig. S2, Table S4b). As the terminal product of diacids photo-oxidation (Kawamura and Sakaguchi, 1999),  $C_2$  also displayed marked day-night differences (daytime:  $766 \pm 552$  ng m-3 vs. nighttime:  $585 \pm 497$  ng m-3, p = 0.023), reflecting strong anthropogenic influence on ground-level photochemistry. In contrast,  $C_2$  at the top of the Mount Hua (daytime:  $312 \pm 224$  ng m-3 vs. nighttime:  $299 \pm 186$  ng m-3; p = 0.941) and its precursors showed no clear

day-night differences (p = 0.341–0.917; Table S4c), consistent with the patterns of PM2.5 (p = 0.979), OC (p = 0.766), and other major components (Table S4a).

Table S3a. Independent samples t-test results for  $PM_{2.5}$ , major chemical components, and meteorological parameters between the foot and top of Mount Hua during non-dust events

| Parameter                               | Mean difference
(Foot-top) | 95% CI       | t-value | df     | p-value |
|-----------------------------------------|-------------------------------|--------------|---------|--------|---------|
| PM 2.5 (μg m -3 ) | 99                            | [86, 113]    | 14.60   | 61.05  | <0.001  |
| $OC (\mu g m^{-3})$                     | 12                            | [10, 15]     | 10.83   | 60.59  | <0.001  |
| EC ( $\mu g m^{-3}$ )                   | 3.6                           | [3.1, 4.2]   | 12.44   | 57.00  | <0.001  |
| WSOC (µg m -3 )              | 6.2                           | [4.5, 7.9]   | 7.34    | 64.41  | <0.001  |
| $SO_4^{2-}$ (µg m -3 )       | 5.2                           | [4.0, 6.4]   | 8.40    | 64.57  | <0.001  |
| $NO_{3}^{-}$ (µg m -3 )      | 12                            | [8.3, 16]    | 6.38    | 65.11  | <0.001  |
| Cl - (µg m -3 )   | 3.0                           | [2.6, 3.4]   | 13.51   | 55.09  | <0.001  |
| $Ca^{2+} (\mu g m^{-3})$                | 1.3                           | [1.0, 1.7]   | 7.06    | 94.60  | <0.001  |
| $K^+$ (µg m -3 )             | 0.6                           | [0.4, 0.7]   | 9.96    | 57.41  | <0.001  |
| ${ m Mg^{2+}}({ m \mu g}\;{ m m^{-3}})$ | 0.05                          | [0.03, 0.08] | 4.12    | 102.3  | <0.001  |
| $NH_4^+ (\mu g m^{-3})$                 | 1.4                           | [1.1, 1.7]   | 8.07    | 82.95  | <0.001  |
| $O_3 (\mu g m^{-3})$                    | -22                           | [-28, -16]   | -6.94   | 90.69  | <0.001  |
| T (°C)                                  | 5.9                           | [3.8, 7.9]   | 5.60    | 92.06  | <0.001  |
| RH (%)                                  | 7.6                           | [0.8, 14]    | 2.23    | 103.71 | 0.028   |
| WS (m s -1 )                 | -4.1                          | [-5.0, -3.3] | -9.69   | 56.94  | <0.001  |

Table S3b. Independent samples t-test results of dicarboxylic acids and related compounds in  $PM_{2.5}$  at the foot and top of Mount Hua during non-dust events

| Compound                        | Mean difference
(Foot-top) | 95% CI        | t-value | df     | p-value |
|---------------------------------|-------------------------------|---------------|---------|--------|---------|
| Oxalic, C 2          | 368                           | [213, 523]    | 4.74    | 67.16  | <0.001  |
| Malonic, C 3         | 25                            | [6.6, 43]     | 2.71    | 87.37  | 0.008   |
| Succinic, C 4        | 73                            | [48, 99]      | 5.70    | 82.10  | <0.001  |
| Glutaric, C 5        | 12                            | [4.5, 20]     | 3.16    | 98.72  | 0.002   |
| Adipic, C 6          | 4.6                           | [-0.001, 9.3] | 1.99    | 93.82  | 0.05    |
| Pimelic, C 7         | 4.4                           | [2.8, 6.0]    | 5.48    | 81.2   | <0.001  |
| Suberic, C 8         | 3.1                           | [1.7, 4.5]    | 4.28    | 103.6  | <0.001  |
| Azelaic, C9                     | 126                           | [95, 156]     | 8.24    | 54.77  | <0.001  |
| Sebacic, C 10        | 11                            | [7.2, 15]     | 5.79    | 59.20  | <0.001  |
| Undecanedioic, C 11  | 3.3                           | [2.3, 4.3]    | 6.40    | 79.53  | <0.001  |
| Methylmalonic, iC 4  | 17                            | [12, 22]      | 6.81    | 78.91  | <0.001  |
| Mehtylsuccinic, iC5             | 5.8                           | [3.8, 7.8]    | 5.83    | 69.66  | <0.001  |
| Methylglutaric, iC 6 | -3.6                          | [-6.1, -1.0]  | -2.77   | 103.13 | 0.007   |
| Maleic, M                       | 6.1                           | [2.2, 10]     | 3.11    | 90.69  | 0.002   |
| Fumaric, F                      | 2.2                           | [1.3, 3.1]    | 5.01    | 66.42  | <0.001  |
| Methylmaleic, mM                | 12                            | [4.8, 19]     | 3.39    | 62.64  | 0.001   |
| Phthalic, Ph                    | 39                            | [25, 52]      | 5.72    | 75.35  | <0.001  |
| Isophthalic, iPh                | 7.4                           | [5.7, 9.2]    | 8.47    | 54.78  | <0.001  |
| Ketopimelic, kC7                | 1.7                           | [0.9, 2.4]    | 4.29    | 99.63  | <0.001  |
| Pyruvic, Pyr                    | 26                            | [9.1, 43]     | 3.05    | 88.86  | 0.003   |
| Glyoxylic, ωC 2      | 137                           | [80, 193]     | 4.84    | 59.09  | <0.001  |
| Glyoxal, Gly                    | 4.7                           | [-1.5, 11]    | 1.52    | 84.57  | 0.133   |
| Methylglyoxal, mGly             | 59                            | [44, 74]      | 7.72    | 77.83  | <0.001  |
| Benzoic                         | 4.0                           | [1.6, 6.3]    | 3.34    | 96.15  | 0.001   |
| Category Summary                |                               |               |         |        |         |
| Dicarboxylic acids              | 718                           | [453, 983]    | 5.40    | 69.15  | <0.001  |
| Ketocarboxylic acids            | 163                           | [91, 234]     | 4.53    | 63.76  | <0.001  |
| α-Dicarbonyls                   | 63                            | [44, 83]      | 6.48    | 82.19  | <0.001  |

Note: All concentrations are in ng m-3.

Table S4a. Paired samples t-test results of day-night differences for PM2.5, major chemical components, and meteorological parameters at the foot and top of Mount Hua during non-dust events

| Parameter                             | Sites | Mean difference (Day-night) | 95% CI        | t-value | p-value |
|---------------------------------------|-------|-----------------------------|---------------|---------|---------|
| D) ( 2)                               | Foot  | 24                          | [4.0, 45]     | 2.46    | 0.021   |
| $PM_{2.5} (\mu g m^{-3})$             | Top   | -0.09                       | [-7.4, 7.2]   | -0.026  | 0.979   |
| 000 ( -2)                             | Foot  | 4.4                         | [1.4, 7.4]    | 2.99    | 0.006   |
| $OC (\mu g m^{-3})$                   | Top   | -0.1                        | [-1.0, 0.7]   | -0.30   | 0.766   |
|                                       | Foot  | 1.5                         | [0.7, 2.3]    | 3.96    | 0.001   |
| EC ( $\mu g m^{-3}$ )                 | Top   | 0.03                        | [-0.2, 0.2]   | 0.27    | 0.793   |
|                                       | Foot  | 2.7                         | [0.7, 4.6]    | 2.84    | 0.009   |
| WSOC (μg m -3 )            | Top   | 0.1                         | [-0.6, 0.9]   | 0.30    | 0770    |
|                                       | Foot  | 1.7                         | [0.1, 3.2]    | 2.23    | 0.035   |
| $SO_4^{2-}$ (µg m -3 )     | Top   | 0.3                         | [-0.2, 0.9]   | 1.37    | 0.184   |
|                                       | Foot  | 2.6                         | [-1.1, 6.3]   | 1.44    | 0.163   |
| $NO_{3}^{-}$ (µg m -3 )    | Top   | 0.2                         | [-2.0, 2.3]   | 0.16    | 0.878   |
|                                       | Foot  | 0.6                         | [0.2, 1.0]    | 2.89    | 0.008   |
| Cl - (µg m -3 ) | Top   | 0.1                         | [-0.03, 0.20] | 1.59    | 0.123   |
|                                       | Foot  | 0.8                         | [0.4, 1.2]    | 3.78    | 0.001   |
| $Ca^{2+}$ (µg m -3 )       | Top   | 0.03                        | [-0.4, 0.4]   | -0.14   | 0.889   |
|                                       | Foot  | 0.2                         | [0.1, 0.3]    | 3.31    | 0.003   |
| $K^+$ (µg m -3 )           | Top   | -0.01                       | [-0.05, 0.03] | -0.39   | 0.703   |
|                                       | Foot  | 0.04                        | [0.01, 0.07]  | 3.05    | 0.005   |
| $Mg^{2+}$ (µg m -3 )       | Top   | 0.01                        | [-0.02, 0.04] | -0.53   | 0.602   |
|                                       | Foot  | 0.8                         | [-0.3, 0.4]   | 0.47    | 0.640   |
| $\mathrm{NH_4^+}(\mu g\;m^{-3})$      | Top   | -0.02                       | [-0.2, 0.2]   | -0.16   | 0.875   |
|                                       | Foot  | 18                          | [11, 26]      | 4.95    | <0.001  |
| $O_3 (\mu g m^{-3})$                  | Top   | 1.9                         | [0.4, 3.5]    | 2.53    | 0.019   |
|                                       | Foot  | 4.2                         | [3.1, 5.4]    | 7.34    | <0.001  |
| T (°C)                                | Top   | 1.8                         | [0.6, 3.0]    | 3.00    | 0.006   |
|                                       | Foot  | -18                         | [-21, -14]    | -10.89  | <0.001  |
| RH (%)                                | Top   | -6.0                        | [-11, -0.9]   | -2.40   | 0.024   |
|                                       | Foot  | 0.4                         | [0.1, 0.7]    | 2.60    | 0.015   |
| WS (m s -1 )               | Top   | -0.8                        | [-2.1, 0.4]   | -1.34   | 0.192   |

Table S4b. Paired samples t-test results for day-night differences of dicarboxylic acids and related compounds in  $PM_{2.5}$  at the foot of Mount Hua during non-dust events

| Compound                        | Mean difference
(Foot-top) | 95% CI      | t-value | p-value |
|---------------------------------|-------------------------------|-------------|---------|---------|
| Oxalic, C 2          | 154                           | [23, 284]   | 2.41    | 0.023   |
| Malonic, C 3         | 12                            | [-2.3, 26]  | 1.71    | 0.098   |
| Succinic, C 4        | 34                            | [11, 58]    | 3.02    | 0.006   |
| Glutaric, C 5        | 10                            | [4.1, 16]   | 3.50    | 0.002   |
| Adipic, C 6          | 2.0                           | [-2.1, 6.1] | 1.00    | 0.326   |
| Pimelic, C 7         | 1.8                           | [0.2, 3.6]  | 2.27    | 0.031   |
| Suberic, C 8         | 1.4                           | [0.01, 2.8] | 2.07    | 0.049   |
| Azelaic, C9                     | 38                            | [-2.0, 77]  | 1.95    | 0.061   |
| Sebacic, C 10        | 4.0                           | [-1.3, 9.3] | 1.56    | 0.132   |
| Undecanedioic, C 11  | 1.4                           | [-0.6, 3.4] | 1.41    | 0.171   |
| Methylmalonic, iC 4  | 11                            | [4.9, 17]   | 3.77    | 0.001   |
| Mehtylsuccinic, iC 5 | -1.1                          | [-4.0, 1.8] | -0.75   | 0.459   |
| Methylglutaric, iC 6 | -3.0                          | [0.6, 5.5]  | 2.52    | 0.018   |
| Maleic, M                       | 3.8                           | [0.1, 7.4]  | 2.11    | 0.044   |
| Fumaric, F                      | 0.6                           | [-1.0, 2.3] | 0.81    | 0.426   |
| Methylmaleic, mM                | 1.0                           | [-6.0, 8.0] | 0.29    | 0.776   |
| Phthalic, Ph                    | 15                            | [-3.4, 33]  | 1.68    | 0.106   |
| Isophthalic, iPh                | 1.3                           | [-1.1, 3.7] | 1.09    | 0.287   |
| Ketopimelic, kC7                | 0.4                           | [-0.6, 1.4] | 0.84    | 0.409   |
| Pyruvic, Pyr                    | 8.4                           | [-9.4, 26]  | 0.97    | 0.341   |
| Glyoxylic, ωC 2      | 23                            | [-38, 84]   | 0.77    | 0.448   |
| Glyoxal, Gly                    | -1.0                          | [-8.8, 6.9] | -0.26   | 0.797   |
| Methylglyoxal, mGly             | 36                            | [18, 53]    | 4.13    | <0.001  |
| Benzoic                         | 1.7                           | [-1.0, 4.4] | 1.33    | 0.196   |
| Category Summary                |                               |             |         |         |
| Dicarboxylic acids              | 293                           | [50, 536]   | 2.47    | 0.020   |
| Ketocarboxylic acids            | 31                            | [-45, 107]  | 0.85    | 0.404   |
| α-Dicarbonyls                   | 35                            | [13, 56]    | 3.33    | 0.003   |

Note: All concentrations are in ng m-3.

Table S4c. Paired samples t-test results for day-night differences of dicarboxylic acids and related compounds in  $PM_{2.5}$  at the top of Mount Hua during non-dust events

| Compound                        | Mean difference
(Foot-top) | 95% CI      | t-value | p-value |
|---------------------------------|-------------------------------|-------------|---------|---------|
| Oxalic, C 2          | 1.9                           | [-51, 55]   | 0.075   | 0.941   |
| Malonic, C 3         | -1.2                          | [-13, 10]   | -0.22   | 0.828   |
| Succinic, C 4        | 0.01                          | [-12, 12]   | 0.002   | 0.998   |
| Glutaric, C 5        | 3.0                           | [-4.1, 10]  | 0.87    | 0.390   |
| Adipic, C 6          | 1.4                           | [-2.4, 5.3] | 0.78    | 0.443   |
| Pimelic, C 7         | -0.1                          | [-1.1, 0.9] | -0.13   | 0.897   |
| Suberic, C 8         | 0.5                           | [-1.2, 2.2] | 0.61    | 0.550   |
| Azelaic, C9                     | 0.2                           | [-6.1, 6.5] | 0.062   | 0.951   |
| Sebacic, C 10        | -0.2                          | [-1.3, 0.9] | -0.36   | 0.720   |
| Undecanedioic, C 11  | 0.4                           | [-0.6, 1.3] | 0.76    | 0.457   |
| Methylmalonic, iC 4  | 0.4                           | [-2.6, 3.4] | 0.28    | 0.782   |
| Mehtylsuccinic, iC 5 | -0.1                          | [-1.7, 1.5] | -0.14   | 0.892   |
| Methylglutaric, iC 6 | -0.6                          | [-3.9, 2.8] | -0.35   | 0.733   |
| Maleic, M                       | 0.0                           | [-3.3, 3.3] | 0.00    | 1.000   |
| Fumaric, F                      | -0.1                          | [-1.0, 0.7] | -0.30   | 0.770   |
| Methylmaleic, mM                | -0.4                          | [-3.2, 2.5] | -0.27   | 0.792   |
| Phthalic, Ph                    | 3.4                           | [-2.9, 9.8] | 1.12    | 0.274   |
| Isophthalic, iPh                | -0.2                          | [-0.9, 0.4] | -0.71   | 0.481   |
| Ketopimelic, kC7                | 0.004                         | [-0.7, 0.7] | 0.011   | 0.992   |
| Pyruvic, Pyr                    | -3.2                          | [-14, 7.8]  | -0.60   | 0.553   |
| Glyoxylic, ωC 2      | 1.4                           | [-12, 15]   | 0.21    | 0.834   |
| Glyoxal, Gly                    | -1.8                          | [-5.5, 2.0] | -0.97   | 0.341   |
| Methylglyoxal, mGly             | -0.5                          | [-9.2, 8.3] | -0.11   | 0.917   |
| Benzoic                         | -0.1                          | [-2.0, 1.9] | -0.075  | 0.941   |
| Category Summary         |                               |             |         |         |
| Dicarboxylic acids              | 17                            | [-87, 122]  | -0.343  | 0.734   |
| Ketocarboxylic acids            | -0.9                          | [-24, 22]   | -0.082  | 0.935   |
| α-Dicarbonyls                   | -1.4                          | [-13, 11]   | -0.24   | 0.813   |

Note: All concentrations are in ng m-3.

3. Introduction section — Scientific motivation for foot vs. top comparison. You reference prior work (Shen et al., 2023) that already found distinct C2 formation pathways at different elevations of Mount Hua during the summer. Please state explicitly: (i) the knowledge gap left by your previous studies, (ii) what new question or hypothesis this study addresses beyond the prior work.

**Response:** We thank the reviewer for this critical comment. We have revised the Introduction to explicitly state the knowledge gap remaining after our previous study (Shen et al., 2023) and the new scientific hypothesis addressed in this work. The modified text now reads as follows:

Tropospheric aerosols in high mountain areas are significantly influenced by long-range transport of surface pollutants, making them more representative of regional atmospheric quality. Our previous study (Shen et al., 2023) demonstrated that summer daytime valley winds on Mount Hua transport organic acids from the foot to top, thereby altering the chemical composition of the free troposphere and establishing distinct formation pathways of C2 at different altitudes. This study examines aerosol vertical distribution characteristics in winter. Low temperatures cause a significant reduction in the boundary layer height over Mount Hua, which inhibits the diffusion of local pollutants to the top. Consequently, the top remains in a free tropospheric environment where aerosols originate primarily from long-range transport from dust source regions (Liu et al., 2024). During the observational period, we documented a major dust event in which particles from dust source regions were directly transported to the top. There, they mixed with local anthropogenic pollutants, triggering complex atmospheric chemical reactions that resulted in notable vertical differences in aerosol chemical properties. However, the influence of heterogeneous reactions on dust aerosol surfaces on the generation of organic acids, particularly their role in modifying C2 formation mechanisms at different altitudes remains unclear. Therefore, using observational data from a typical dust event in winter 2021, this study focuses on examining the impacts of dust transport on the molecular distribution, particle size characteristics of diacids, and the formation mechanisms of C2. The work aims to elucidate the key role of heterogeneous chemical reactions on the surfaces of dust aerosols in the formation of SOA, providing new insights into regional atmospheric chemical processes during dust events.

4. 173–179 — Interpretation of diurnal behavior at altitude. The statement that oxalic acid (C2) and its precursors show no significant diurnal fluctuation at the summit, implying limited local photochemical contributions aloft, is presented in the manuscript without supporting references or mechanistic discussion. This interpretation seems to unclear and potentially inconsistent with established understanding. Cloud and in-cloud processes are widely recognized as important pathways for aqSOA formation in the free troposphere, and oxalate is known to associate preferentially with Fe-containing particles during in-cloud processing (e.g., Zhang et al., 2019, DOI: 10.1021/acs.est.8b05280). Fe is also known to catalyze the photochemical decomposition of oxalic acid. In this context, the claim that high-altitude C2 shows no significant diurnal variation requires further justification and supporting references, as it appears difficult to reconcile with existing literature.

**Response:** We thank the reviewer for their valuable comments. In response to the points raised, we have made the following revisions to the manuscript. First, to ensure

terminological accuracy, we have systematically replaced "diurnal variation" with "day-night differences" throughout the text. This revision better aligns with our defined sampling periods, daytime (08:00–19:00) and nighttime (20:00–07:00 the next day). Second, the absence of clear day-night differences is consistent with the environmental characteristics of the high-altitude free troposphere. Such stability is typical for high-altitude sites located above the planetary boundary layer and is primarily governed by regional transport processes (Fu et al., 2010; Li et al., 2012; Meng et al., 2014). We fully agree with the reviewer's insight regarding the importance of in-cloud processes and have accordingly expanded the relevant section in the manuscript. The revised text now emphasizes that in-cloud processes including the in-cloud oxidation of precursors and the iron-catalyzed decomposition of C2 represent key mechanisms regulating the stable distribution of organic acid concentrations at high altitudes. Supporting references, such as Zhang et al. (2019), have been added accordingly.

Day-night differences provide further evidence supporting this conclusion. At the foot of the mountain, PM2.5, ionic components, and carbonaceous components (except for NO3- and  $NH_4^+$ ) all showed significant day-night differences (p < 0.05; Table S4a). Among the diacids, methylglyoxal (mGly) exhibited the most pronounced day-night differences (daytime:  $128 \pm$ 49 ng m-3, 47% higher than nighttime:  $87 \pm 42$  ng m-3, p < 0.001; Fig. S2, Table S4b). As the terminal product of diacids photo-oxidation (Kawamura and Sakaguchi, 1999), C2 also displayed marked day-night differences (daytime:  $766 \pm 552$  ng m-3 vs. nighttime:  $585 \pm 497$ ng  $m^{-3}$ , p = 0.023), reflecting strong anthropogenic influence on ground-level photochemistry. In contrast,  $C_2$  at the top of the Mount Hua (daytime:  $312 \pm 224$  ng m-3 vs. nighttime:  $299 \pm$ 186 ng m-3; p = 0.941) and its precursors showed no clear day-night differences (p = 0.341-0.917; Table S4c), consistent with the patterns of PM2.5 (p = 0.979), OC (p = 0.766), and other major components (Table S4a). Such stability is a typical characteristic of high-altitude sites located above the planetary boundary layer, primarily governed by regional transport processes (Fu et al., 2008; Li et al., 2012; Meng et al., 2014), indicating that aerosol processes in the free troposphere differ from those at ground level. In this high-altitude environment, in-cloud processes represent a key pathway for aqSOA formation. Studies have shown that C2 mainly originates from the in-cloud oxidation of precursors such as glyoxal (Gly) and isoprene (Warneck, 2003; Lim et al., 2005; Carlton et al., 2006), a mechanism supported by global model simulations (Myriokefalitakis et al., 2011). Additionally, the photochemical decomposition of C2 following its association with Fe-containing particles in clouds (Zhang et al., 2019) also contributes to the stable distribution of organic acid concentrations at high altitudes.

5. 241: You attribute higher OC fraction to "enhanced photochemical reactions and gas-particle conversion," but no direct evidence is shown. Earlier text in your manuscript also suggests mixing with anthropogenic carbonaceous aerosols during long-range transport, and you report elevated azelaic acid (biomass burning tracer) at the summit. The manuscript does not provide clear evidence to distinguish between secondary formation and primary biomass-burning/transport contributions as explanations for the higher OC fraction.

**Response:** We thank the reviewer for this critical comment. We agree that attributing the increase in the OC fraction specifically to "enhanced photochemical reactions and gas-particle conversion" lacked direct evidence and was speculative. Therefore, we have removed this

statement from the manuscript. In the revised version, we have retained the description of "efficient mixing of dust with anthropogenic carbonaceous aerosols during long-range transport, coupled with more vigorous secondary formation processes in the free troposphere." We believe this statement is a reasonable interpretation of the observed phenomena, as it is based on actual observational characteristics such as the doubling of OC concentration at the summit and is supported by relevant literature. Furthermore, the concentration peak of the biomass burning tracer C9 did not coincide with the rise in OC, indicating a limited contribution from primary emissions. This evidence further supports the conclusion that transported and secondarily processed aerosols are the main factor responsible for the observed changes in OC.

6. **246:** You ascribe higher SO42- mainly to heterogeneous reactions on dust particle surfaces. An alternative pathway—direct contribution from dust-borne sulfate species such as CaSO4—receives insufficient consideration in the text. The current discussion does not adequately address this alternative or reconcile it with the observations.

**Response:** Thank you for this constructive suggestion. We have revised the discussion to incorporate the potential contribution of dust-borne sulfate species. The updated text in Line 246 now reads as follows:

 $SO_4^{2-}$  concentrations also increased at both sites (foot: 5.8 to 10.0 µg m-3; top: 3.8 to 8.7 µg m-3). This increase can be attributed to both the release of inherent sulfate species in dust (e.g., CaSO4) (Wu et al., 2012) and heterogeneous reactions on dust particle surfaces, where transition metals such as Fe (III) and Mn (II) catalyze the conversion of  $SO_2$  to  $SO_4^{2-}$  (Harris et al., 2013; Myriokefalitakis et al., 2022).

7. 307–310 "The C5 at the foot likely mainly comes from local emission, such as the oxidation of cyclopentene in vehicle exhaust, whereas the high-altitude area is influenced by a combination of long-range transport and local processes. The text does not present the necessary observational support and data analysis to substantiate this source attribution.

**Response:** We thank the reviewer for this critical comment. We agree that the original statement regarding the sources of  $C_5$  was speculative and lacked sufficient supporting evidence. Following the reviewer's suggestion, we have removed the speculative attribution. The revised text now focuses on the objective observational fact by stating: "This difference in modal structure suggests that different atmospheric processes govern the behavior of  $C_5$  at different altitudes."

8. 332–337 — Coarse vs. fine  $C_2$  and the nitrate-coating hypothesis. The manuscript reports a shift from fine- to coarse-mode  $C_2$  during dust events and links this to nitrate coatings formed via  $CaCO_3 + HNO_3 \rightarrow Ca(NO_3)_2$ . However, size-resolved evidence for  $Ca^{2+}$  and  $NO_3$  is not presented in the manuscript, so the causal link to a water-containing nitrate coating is not demonstrated. Relevant literature (e.g., W. J. Li & L. Y. Shao, ACP, 2009; Zhi et al., ES&T, 2025) that discusses the role of Ca-rich coatings during dust aging should be integrated into the discussion.

**Response:** We thank the reviewer for this suggestion. As recommended, we have integrated the relevant literature (Li and Shao, 2009; Zhi et al., 2025) and added our size-resolved ion

measurements (Fig. S5) to support the discussion on nitrate coatings. The revised text now includes this evidence and clarifies the association between coarse-mode C2 and Ca(NO3)2 formation during dust events.

The dust transport process impacted the size distributions of diacids in aerosols (Fig. 5). At the foot of the mountain, the  $C_2$  concentration in fine particles ( $\leq 2.1 \mu m$ ) decreased from 8662 ng m-3 to 4880 ng m-3 (a reduction of 43.7%) during dust events, while the concentration in coarse particle (>2.1 µm) increased from 2718 ng m-3 to 2843 ng m-3 (an increase of 4.6%) (Table 3). A more pronounced change was observed at the high-altitude top of Mount Hua, where the C2 concentration in coarse particles (2301 ng m-3) exceeded that in fine particles (2161 ng m-3) during dust events, indicating a shift in the dominant particle size distribution from fine to coarse modes. This shift in particle size distribution can be attributed to the formation of Ca(NO3)2 coatings resulting from the reaction between calcium carbonate and NO3- during dust aging (Li and Shao, 2009; Zhi et al., 2025). These hygroscopic coatings create favorable conditions for the adsorption and oxidation of gaseous organic compounds, thereby promoting the formation of SOA on the surfaces of coarse particles. Research by Li et al. (2025) further confirms that aqSOA formed on dust surfaces can effectively enhance SOA production and drive a transition in the size distribution from the submicron to the supermicron range, which is highly consistent with the observational results of this study. Size-segregated ion data (Fig. S5) provide direct evidence for the above mechanism. Ca2+ was primarily present in the coarse mode (3.3–5.8 µm), while during dust events, NO3- at the top migrated from the fine mode (0.4-1.1 µm) to the coarse mode (3.3-5.8 µm) and coexisted with Ca2+ in the same size range, strongly supporting the formation of Ca(NO3)2 coatings on dust particle surfaces. In contrast, at the foot of the mountain, although the concentration of NO3- decreased, it remained predominantly in the fine mode. This spatial difference may stem from more thorough aging and reactions of aerosols at the top due to longer transport times. Meanwhile, the foot is influenced by local pollution, resulting in higher background NO3- concentrations and competitive reactions with components such as SO42-, which may collectively delay the distinct shift of NO3- to the coarse mode.

Figure S5 Size distribution of water-soluble inorganic ions at the foot and top of Mount Hua during non-dust and dust periods

**English polishing and wording precision**

9. 49: "secondary substances" → "secondary aerosols".

Response: "secondary substances" has been replaced with "secondary aerosols".

10. 75: "photocatalytic reactions" → "photochemical reactions".

**Response:** The original phrasing has been eliminated as part of a broader revision of the introduction.

11. 163: "The contribution ratios of  $C_9$  at the two sites were 8.5% and 3.2%, respectively"  $\rightarrow$  Specify "contribution to what?" (e.g., total diacids, total organic acids, or OC). Define the denominator.

**Response:** We thank the reviewer for this important clarification. The sentence has been revised to: "Consequently, the contribution of  $C_9$  to total diacids was substantially greater at the foot (8.5%) than at the top (3.2%) (Fig. 2b)".

12. 233: Avoid "modest increase." Provide exact statistics and confidence intervals; define "modest" quantitatively.

**Response:** This is a valid point. We have removed the subjective term "modest" and replaced it with the exact quantitative data: "...increased from 2718 ng m-3 to 2843 ng m-3 (an increase of 4.6%)".

13. 255: "a decrease of 59%"  $\rightarrow$  "decrease by 59%".

**Response:** "a decrease of 59%" has been corrected to "decrease by 59%".

14. 305: "single peak" → "unimodal distribution".

**Response:** "single peak" has been replaced with the more precise "unimodal distribution".

15. 331: "altered the particle size distribution characteristics"  $\rightarrow$  "impacted the size distributions of diacids in aerosols".

**Response:** The text has been revised to "impacted the size distributions of diacids in aerosols".

[revised manuscript text omitted]

---

## Author Response (AR2)

Dear Editor,

We have now revised our manuscript (MS No.: egusphere-2025-3094) according to your final revision request. The abstract and conclusions have been carefully updated in an effort to align with the current ACP style guide. Specifically, we have rewritten the beginning of the abstract to include a brief introduction to the topic for the benefit of non-specialist readers.

Thank you for your guidance and support throughout the review process. Should you have any further questions, please do not he sitate to contact me at <a href="liji@jeecas.cn">liji@jeecas.cn</a>.

Best regards, Jianjun Li On behalf of all co-authors Oct 10, 2025

**Editor comments:**

Public justification (visible to the public if the article is accepted and published):

I have read the reviews and the authors' responses and I am satisfied that the reviewers' comments have been adequately addressed and that the manuscript can go to publication. However before it does, I would ask that the abstract and conclusions be revised so that they conform better to ACP's current style guides: https://www.atmospheric-chemistry-and-physics.net/policies/guidelines\_for\_authors.html. In particular, I would ask that the abstract starts with a brief introduction to the topic and scientific need for the benefit of a nonspecialist reader.

**Response:** We thank the editor for this final guidance. We have thoroughly revised the abstract and conclusions sections to comply with the ACP style guide. Specifically, as requested, we have rewritten the opening of the abstract to provide a clear introduction to the topic and its scientific significance for a nonspecialist audience.

[revised manuscript text omitted]